# An Optical Control Environment for Benchmarking Reinforcement Learning Algorithms

**Abulikemu Abuduweili**                                                  *abulikea@andrew.cmu.edu*
*Robotics Institute, Carnegie Mellon University*

**Changliu Liu**                                                         *cliu6@andrew.cmu.edu*
*Robotics Institute, Carnegie Mellon University*

**Reviewed on OpenReview:** *https://openreview.net/forum?id=61TKzU9B96*

## Abstract

Deep reinforcement learning has the potential to address various scientific problems. In this paper, we implement an optics simulation environment for reinforcement learning based controllers. The environment captures the essence of nonconvexity, nonlinearity, and time-dependent noise inherent in optical systems, offering a more realistic setting. Subsequently, we provide the benchmark results of several reinforcement learning algorithms on the proposed simulation environment. The experimental findings demonstrate the superiority of off-policy reinforcement learning approaches over traditional control algorithms in navigating the intricacies of complex optical control environments.

## 1 Introduction

In recent years, deep reinforcement learning (RL) has been used to solve challenging problems in various fields (Sutton & Barto, 2018), including self-driving car (Bansal et al., 2018) and robot control (Zhang et al., 2015). Among all applications, deep RL made significant progress in playing games on a superhuman level (Mnih et al., 2013; Silver et al., 2014; 2016; Vinyals et al., 2017). Beyond playing games, deep RL has the potential to strongly impact the traditional control and automation tasks in the natural science, such as control problems in chemistry (Dressler et al., 2018), biology (Izawa et al., 2004), quantum physics (Bukov et al., 2018), optics and photonics (Genty et al., 2020).

In optics and photonics, there are particular potentials for RL methods to drive the next generation of optical laser technologies (Genty et al., 2020). That is not only because there are increasing demands for adaptive control and automation (of tuning and control) for optical systems (Baumeister et al., 2018), but also because many phenomena in optics are nonlinear and multidimensional (Shen, 1984), with noise-sensitive dynamics that are extremely challenging to model using conventional approaches. RL methods are able to control multidimensional environments with nonlinear function approximation (Dai et al., 2018). Thus, exploring RL controllers becomes increasingly promising in optics and photonics as well as in scientific research, medicine, and other industries (Genty et al., 2020; Fermann & Hartl, 2013).

In the field of optics and photonics, Stochastic Parallel Gradient Descent (SPGD) algorithm with a PID controller has traditionally been employed to tackle control problems (Cauwenberghs, 1993; Zhou et al., 2009; Abuduweili et al., 2020a). These problems typically involve adjusting system parameters, such as the delay line of mirrors, with the objective of maximizing a reward, such as optical pulse energy. SPGD is a specific case of the stochastic error descent method (Cauwenberghs, 1993; Dembo & Kailath, 1990), which operates based on a model-free distributed learning mechanism. The algorithm updates the parameters by perturbing each individual parameter vector, resulting in a decrease in error or an increase in reward. However, the applicability of SPGD is limited to convex or near-convex problems, while many control problems in optics exhibit non-convex characteristics. As a result, SPGD struggles to find the global optimum of an optics control system unless the initial state of the system is in close proximity to the global optimum. Traditionally,

experts would manually tune the initial state of the optical system, followed by the use of SPGD-PID to control the adjusted system. Nevertheless, acquiring such expert knowledge becomes increasingly challenging as system complexity grows.

To enable efficient control and automation in optical systems, researchers have introduced deep reinforcement learning (RL) techniques (Tünnermann & Shirakawa, 2019; Sun et al., 2020; Abuduweili et al., 2020b; 2021). Previous studies predominantly focused on implementing Deep Q-Network (DQN) (Mnih et al., 2013) and Deep Deterministic Policy Gradient (DDPG) (Lillicrap et al., 2015) in simple optical control systems, aiming to achieve comparable performance to traditional SPGD-PID controllers (Tünnermann & Shirakawa, 2019; Valensise et al., 2021). However, there is a lack of research evaluating a broader range of RL algorithms in more complex optical control environments. The exploration and evaluation of RL algorithms in real-world optical systems pose significant challenges due to the high cost and the need for experienced experts to implement multiple optical systems with various configurations. Even for a simple optical system, substantial efforts and resources are required to instrument and implement RL algorithms effectively.

Simulation has been widely utilized in the fields of robotics and autonomous driving since the early stages of research (Pomerleau, 1998; Bellemare et al., 2013). As the interest and application of learning-based robotics continue to grow, the role of simulation becomes increasingly crucial in driving research advancements. We believe that simulation holds equal importance in evaluating RL algorithms for optical control. However, to the best of our knowledge, there is currently no open-source RL environment available for optical control simulation.

In this paper, we present OPS (Optical Pulse Stacking), an open and scalable simulator designed for controlling typical optical systems. The underlying physics of OPS aligns with various optical applications, including coherent optical inference (Wetzstein et al., 2020) and linear optical sampling (Dorrer et al., 2003), which find applications in precise measurement, industrial manufacturing, and scientific research. A typical optical pulse stacking system involves the direct and symmetrical stacking of input pulses to multiply their energy, resulting in stacked output pulses (Tünnermann & Shirakawa, 2017; Stark et al., 2017; Astrauskas et al., 2017; Yang et al., 2020). By introducing the OPS optical control simulation environment, our objective is to encourage exploration of RL applications in optical control tasks and further investigate RL controllers in natural sciences. We utilize OPS to evaluate several important RL algorithms, including Twin Delayed Deep Deterministic Policy Gradient (TD3) (Fujimoto et al., 2018), Soft Actor-Critic (SAC) (Haarnoja et al., 2018), and Proximal Policy Optimization (PPO) (Schulman et al., 2017). Our findings indicate that in a simple optical control environment (nearly convex), the traditional SPGD-PID controller performs admirably. However, in complex environments (non-convex optimization), SPGD-PID falls short, and RL-trained policies outperform SPGD-PID. Following the reporting of these RL algorithm results, we discuss the potential and challenges associated with RL algorithms in real-world optical systems. By providing the OPS simulation environment and conducting RL algorithm experiments, we aim to facilitate research on RL applications in optics, benefiting both the machine learning and optics communities. The code of the paper is available at `https://github.com/Walleclipse/Reinforcement-Learning-Pulse-Stacking`.

## 2 Simulation environment

### 2.1 Physics of the simulation

The optical pulse stacking (OPS), also known as pulse combination, system employs a recursive approach to stack optical pulses in the time domain. The dynamics of the OPS are similar to the recurrent neural networks (RNN) or Wavenet architecture (Oord et al., 2016). We illustrate the dynamics of the OPS in RNN style as shown in Fig. 1. In the OPS system, the input consists of a periodic pulse train [1] with a repetition period of $T$. Assuming the basic function of the first pulse at time step $t$ is denoted as $E_1 = E(t)$ (a complex function), the subsequent pulses can be described as $E_2 = E(t+T)$, $E_3 = E(t+2T)$, and so on. The OPS system recursively imposes time delays to earlier pulses in consecutive pairs. For instance, in the first stage of OPS, a time-delay controller imposes the delay $\tau_1$ on pulse 1 to allow it to combine (overlap)

---

[1] The periodic pulse train is typically emitted by lasers, where each laser pulse's wave function is nearly identical except for the time delay.

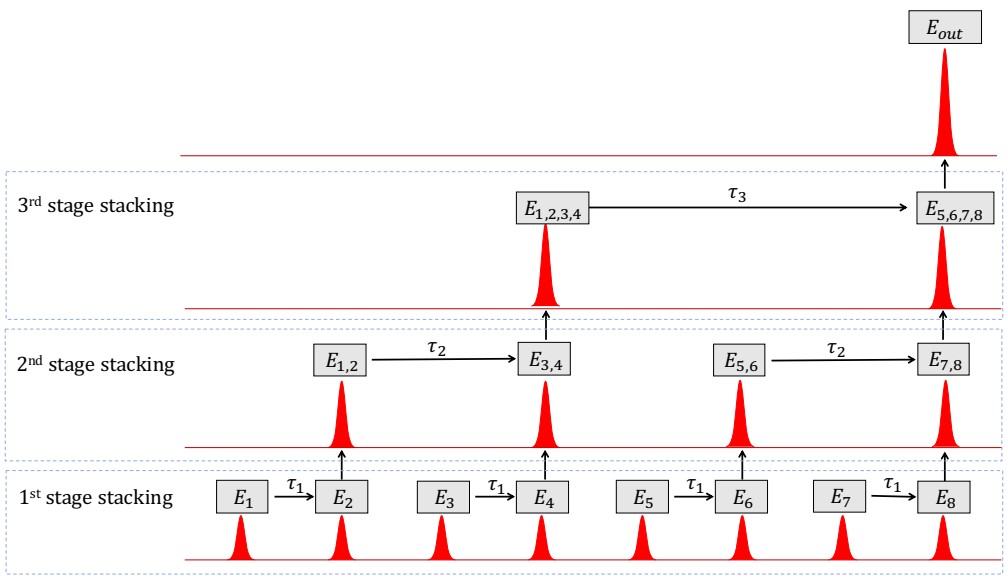

Figure 1: Illustration of the principle of optical pulse stacking. Only 3-stage pulse stacking was plotted for simplicity.

with pulse 2. With the appropriate time delay, pulse 1 can be stacked with the next pulse, $E_2$, resulting in the stacked pulses $E_{1,2} = E(t + \tau_1) + E(t + T)$. Similarly, pulse 3 can be stacked with pulse 4, creating $E_{3,4} = E(t + 2T + \tau_1) + E(t + 3T)$, and so forth. In the second stage of OPS, an additional time delay, $\tau_2$, is imposed on $E_{1,2}$ to allow it to stack with $E_{3,4}$, resulting in $E_{1,2,3,4}$. This stacking process continues in each subsequent stage of the OPS controller, multiplying the pulse energy by a factor of $2^N$ by stacking $2^N$ pulses, where $N$ time delays $(\tau_1, \tau_2, ..., \tau_N)$ are required for control and stabilization. Additional details about the optical pulse stacking are shown in appendix A.1.

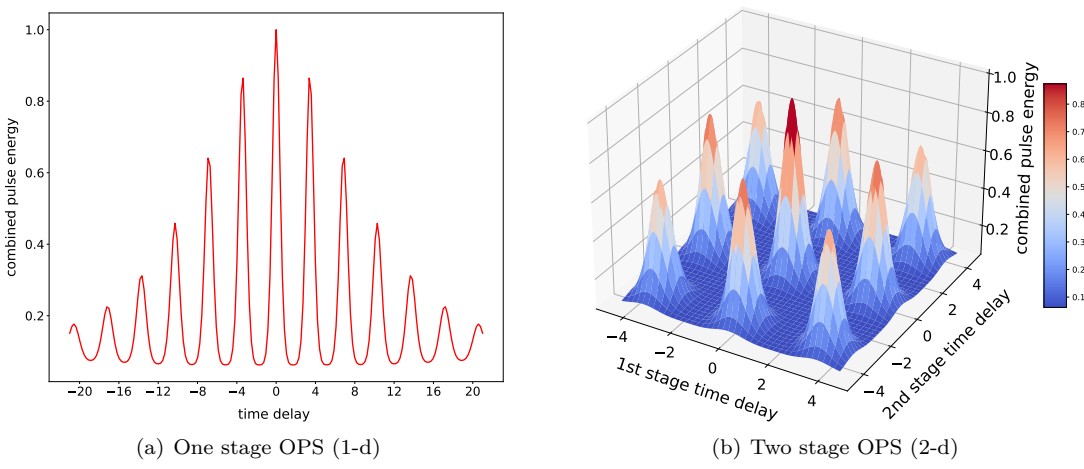

| (a) One stage OPS (1-d) | (b) Two stage OPS (2-d) |

Figure 2: Function of the (a) 1-stage OPS: pulse energy $P_1(\tau_1)$ w.r.t. delay line $\tau_1$. (b) 2-stage OPS: pulse energy $P_2(\tau_1, \tau_2)$ w.r.t. delay lines $(\tau_1, \tau_2)$.

## 2.2 Control objective and noise

The objective of controlling an OPS system is to maximize the energy of the final stacked (output) pulse by adjusting the time delays. We denote the vector of time delays as $\tau = [\tau_1, \tau_2, \cdots, \tau_N]$, and $E_{out}(t; \tau)$ represents the final stacked pulse under the time delay configuration $\tau$. For an $N$-stage OPS system, the energy of the final stacked pulse, denoted as $P_N(\tau)$, is computed as the integral of the norm of $E_{out}(t; \tau)$ over

time. In mathematical terms, we express it as $P_N(\tau) = \int |E_{out}(t;\tau)|dt$. To formulate the objective function for controlling an $N$-stage OPS system, we aim to find the optimal set of time delays $\tau^*$ that maximizes $P_N(\tau)$. The objective function can be defined as follows:

$$\arg\max_\tau P_N(\tau) = \arg\max_{\tau_1,\tau_2,...,\tau_N} P_N(\tau_1, \tau_2, ..., \tau_N) \tag{1}$$

When ignoring noise, the objective function of the final pulse energy, $P_N$, with respect to the time delays, $\tau$, can be derived based on optical coherence. Figure 2(a) depicts the pulse energy function $P_1(\tau_1)$ in a 1-stage OPS system, showing the relationship between pulse energy and the time delay $\tau_1$. Similarly, Fig. 2(b) displays the function surface of $P_2(\tau_1, \tau_2)$ in a 2-stage OPS system, illustrating how pulse energy varies with the first and second stage time delays $(\tau_1, \tau_2)$. As evident from the figures, the control objective of the OPS system is nonlinear and non-convex, even when noise is disregarded. This inherent complexity arises due to factors such as optical periodicity and the nonlinearity of coherent interference. Consequently, achieving the global optimum or better local optima becomes a challenging task for any control algorithms, particularly when starting from a random initial state.

However, in practical scenarios, noise cannot be ignored, and the OPS system is highly sensitive to noise. This sensitivity is primarily due to the pulse wavelength being on the order of micrometers ($1\mu m = 10^{-6}m$). Environmental noise, including vibrations of optical devices and atmospheric temperature drift, can easily cause shifts in the time delays, resulting in changes to the output pulses. As a result, the objective function in real-world applications is much more complex than what is depicted in Fig. 2, especially for higher-stage OPS systems with higher dimensions. Consequently, achieving the control objective becomes even more challenging in the presence of unknown initial states and unpredictable noise in such noise-sensitive complex systems (Genty et al., 2020). Therefore, model-based controllers face significant difficulties in implementation. In this paper, we primarily focus on model-free reinforcement learning approaches to address these challenges.

In this simulation, we incorporate two types of noise: fast noise arising from device vibrations and slow noise caused by temperature drift. The fast noise is modeled as a zero-mean Gaussian random noise with variance $\sigma^2$, following the simulation noise approach outlined in (Tünnermann & Shirakawa, 2019). On the other hand, the slow noise $\mu_t$ accounts for temperature drift and is represented as a piecewise linear function (Ivanova et al., 2021). To capture the combined effect of these two noise sources, we define the overall noise $e_t$ as a random process. Specifically, we can express it as follows:

$$\mathbb{E}[e_t] = \mu_t, \quad \mathbb{VAR}[e_t] = \sigma^2 \tag{2}$$

### 2.3 Reinforcement learning environment

**Interactions with RL agent.** An RL agent interacts with the OPS environment in discrete time steps, as shown in Fig. 3. At each time step $t$, the RL agent receives the current state of the OPS environment, denoted as $s_t$. Based on this state, the agent selects an action $a_t$ to be applied to the environment. The action could involve adjusting the time delays $\tau$ in the OPS system. Once the action is chosen, it is transmitted to the OPS environment, which then processes the action and transitions to a new state $s_{t+1}$. The OPS environment provides feedback to the RL agent in the form of a reward $r_t$. The reward serves as a measure of how well the OPS system is achieving the objective of maximizing the final stacked pulse energy. The RL agent utilizes the experience tuple $(s_t, a_t, s_{t+1}, r_t)$ to learn and update its policy $\pi(a,s)$ over time. The goal of the agent is to learn a policy that maximizes the expected cumulative reward over the interaction with the OPS environment.

**State space.** The state space of the OPS system is a continuous and multidimensional vector space. The state value at time step $t$, denoted as $s_t$, corresponds to the pulse amplitude measurement of the final stacked pulse, given by $s_t = |E_{out}(t;\tau)|$. Therefore, $s_t$ provides a time-domain representation of the final stacked pulse, offering direct insight into the control performance. In practical implementations, the pulse amplitude is typically detected using a photo-detector and subsequently converted into digital time-series signals. In our simulation, we have incorporated real-time rendering of the pulse amplitude to facilitate the monitoring of the control process.

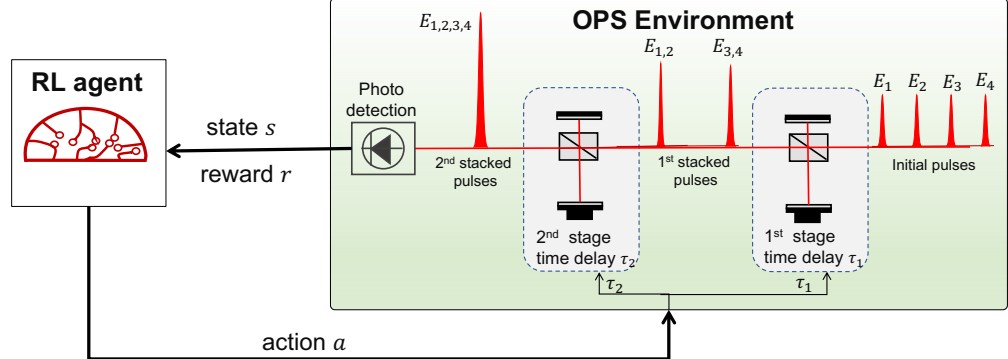

Figure 3: Illustration of the interaction between RL agent and OPS environment. Only 2-stage OPS was plotted for simplicity.

**Action space.** The action space of an $N$-stage OPS environment is a continuous and $N$-dimensional vector space. At each time step $t$, the action $a_t$ corresponds to an additive time delay value $\Delta\tau(t)$ for the $N$-stage OPS environment: $a_t = \Delta\tau(t) = \tau(t+1) - \tau(t)$. The OPS environment applies the additive time delay value $a(t)$ to transition to the next state.

**Reward.** As mentioned in section 2.2, the objective of the OPS controller is maximizing the final stacked pulse energy $P_N(\tau)$. In our simulation, we use the normalized final pulse energy as the reward value. The reward at each time step is defined as:

$$r = -\frac{(P_N(\tau) - P_{max})^2}{(P_{min} - P_{max})^2}, \tag{3}$$

where $P_{max}$ is the maximum pulse energy at the global optimum, and $P_{min}$ is the minimum pulse energy. The maximum reward 0 achieved when $P(\tau) = P_{max}$ (peak position of Fig. 2(b)) .

**State transition function.** The environmental noise has direct impacts on the delay lines, including the vibration and temperature-induced shift noise of the delay line devices. Therefore, in the state transition process, the actual applied delay line value $\tau_{\text{real}}(t + 1)$ is a combination of the action $a_t$ and the noise $e_t$. Specifically, it can be expressed as:

$$\tau_{\text{real}}(t + 1) = \tau_{\text{real}}(t) + a_t + e_t. \tag{4}$$

After selecting the pulses, the real-time delay $\tau_{\text{real}}(t + 1)$ is imposed on them using delay line devices, which introduce additional time delays for the pulses. The state transition process is governed by the combination of the current state, the action taken, and the noise present. The specific form of the state transition follows the principles of coherent light interference (Saleh & Teich, 2019). Let $f$ be an interference observation function. Then the state transition can be written as:

$$s_{t+1} = f(\tau_{\text{real}}(t + 1)) = f(\tau_{\text{real}}(t) + a_t + e_t) = f(f^{-1}(s_t) + a_t + e_t) \tag{5}$$

It is important to note that the slow-changing noise term $\mathbb{E}[e_t] = \mu_t$ follows a piecewise linear function that changes slowly over time. During episodic training for RL agents, $\mu_t$ can be treated as a constant value within an episode. However, the value of $\mu_t$ may vary from one episode to the next. In this case, assuming that $\mu_t$ changes very slowly, the OPS control process can be modeled as a Markov decision process (MDP).

| Mode | Initial state | Noise | Objective |
|---|---|---|---|
| Easy | near the optimum | time-independent: $\frac{d\mu_t}{dt} \equiv 0$ | convex |
| Medium | random | time-independent: $\frac{d\mu_t}{dt} \equiv 0$ | non-convex |
| Hard | random | time-dependent: $\frac{d\mu_t}{dt} \not\equiv 0$ | non-convex |

Table 1: Different difficulty modes on OPS.

```python
from optics_env import OPS_env
env = OPS_env(stage=5, mode="medium")
env.reset()
done = False
while not done:
    action = env.action_space.sample()
    observation, reward, done, info = env.step(action)
    env.render()
```

Figure 4: Example code of the OPS environment.

**Control difficulty of the environment.** We have implemented the OPS environment to support arbitrary stages of pulse stacking ($N \in 1, 2, 3, ...$). As the number of stages increases, the control task becomes more challenging. In addition to the customizable number of stages, we have also introduced three difficulty modes (easy, medium, and hard) for each stage of OPS, as outlined in table 1. The difficulty mode is determined by the initial state of the system and the distribution of noise. These difficulty modes allow for different levels of complexity and challenge in the control task, providing flexibility for evaluating and training control algorithms in various scenarios.

- Easy mode. In the easy mode of the OPS environment, the initial state of the system is set to be near the global optimum. This configuration is often encountered in traditional optics control problems where experts fine-tune the initial state to facilitate easier control. As depicted in Fig. 5(a), we provide an example of the initial state for the easy mode in a 3-stage OPS environment. The proximity of the initial state to the global optimum allows the control objective in the easy mode to be considered convex.

- Medium mode. In the medium mode of the OPS environment, the initial state of the system is randomly determined, as illustrated in Fig. 5(b). This random initialization introduces non-convexity into the control problem, making it more challenging to solve. However, in the medium mode, the noise present in the system is time-independent. We model this noise as a Gaussian distribution, where $e_t$ follows a normal distribution $\mathcal{N}(\mu, \sigma)$. This setting aligns with classical reinforcement learning and typical Markov Decision Process (MDP) settings, where the noise distribution remains the same throughout each episode.

- Hard mode. In the hard mode of the OPS environment, similar to the medium mode, the initial state of the system is randomly determined. However, in contrast to the medium mode, the behavior of the noise in the hard mode is more complex. The mean value of the noise distribution $\mu_t$ becomes a time-dependent variable that slowly changes over time. This time-dependent noise introduces additional challenges and complexity to the control problem. In this case, the control problem deviates from a typical Markov Decision Process (MDP) setting. By incorporating the noise term into the state definition as $\hat{s}_t = [s_t; e_t]$, we effectively convert the control process into a Partially Observable Markov Decision Process (POMDP). The hard mode is designed to mimic real-world settings more closely. In practical applications, when deploying the trained model in a testing environment, we often encounter temperature drift, which causes the noise distribution of the testing environment to differ from the training environment. Therefore, the hard mode simulates the realistic scenario where the noise distribution is not stationary and may vary over time, making the control problem more challenging and closer to real-world conditions.

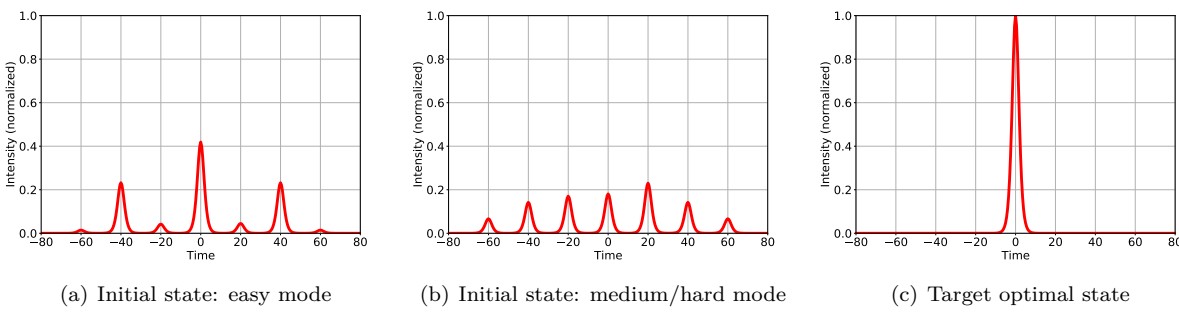

(a) Initial state: easy mode     (b) Initial state: medium/hard mode     (c) Target optimal state

Figure 5: Rendering examples of the (a) initial state for easy mode, (b) initial state for medium or hard mode, (c) global optimal target state in a 2-stage OPS. In the initial state of the easy mode is closer to the target state. The initial state of medium or hard mode is random.

**API & sample usage.** The simulation in this study is based on the Nonlinear-Optical-Modeling (Hult, 2007), which provides the optical and physical fundamentals for the OPS environment. To facilitate integration and compatibility with existing frameworks and tools, the OPS environment is designed to be compatible with the widely used OpenAI Gym API (Brockman et al., 2016). To demonstrate the usage of the OPS environment, we provide an example code snippet in Fig. 4.

**Features of the OPS environment.** We summarize the key features of the OPS environment as follows:

- Open-source optical control environment: To the best of our knowledge, this is the first open-sourced RL environment for optical control problems. The use of open-source licenses enables researchers to inspect the underlying code and modify the environment if necessary to test new research ideas.

- Scalable and difficulty-adjustable scientific environment: Unlike many RL environments that are easy to solve, our OPS environment allows flexible adjustment of difficulty. The dimension of the action space can easily scale with the stage number $N$. Choosing a larger $N$ with the hard mode makes controlling the environment more challenging. Effective solutions to hard scientific control problems can have a broad impact on various scientific control problems (Genty et al., 2020; Fermann & Hartl, 2013).

- Realistic noise: In the hard mode of the OPS environment, we model the noise distribution as a time-dependent function. This mirrors a real-world scenario in which a noise-distribution shift occurs between the testing and training environments. Such realistic noise modeling is particularly relevant for noise-sensitive systems (Ivanova et al., 2021) and increases the stochasticity of the environment.

- Extendable state and structural information: When $\mu_t$ changes very slowly, the OPS control process can be formulated as an MDP. In cases where $\mu_t$ undergoes substantial variations and the noise $e_t$ varies with time, the inclusion of the noise term within the state definition as $\hat{s}_t = [s_t; e_t]$ seamlessly transforms the control process into a POMDP. Furthermore, we can explore the structural information or physical constraints from the function of the OPS (see Fig. 2) and incorporate it with RL controllers.

## 3 Experiments

### 3.1 Experimental setup

We present benchmark results for various control algorithms, including a traditional control algorithm and three state-of-the-art reinforcement learning algorithms:

- **SPGD** (Stochastic Parallel Gradient Descent) algorithm with PID controller: SPGD is a widely used approach for controlling optical systems (Cauwenberghs, 1993; Zhou et al., 2009). In SPGD, the objective gradient estimation is achieved by applying a random perturbation to the time delay value, denoted as $\delta\tau$. The update formula is $\tau(t+1) = \tau(t) + \eta[P_N(\tau(t)+\delta\tau) - P_N(\tau(t))]\delta\tau$, where $\eta$ is the update step-size. The output of the SPGD algorithm is then sent to a PID controller to control the system. In this work, we refer to the SPGD-PID controller as the SPGD controller.

- **PPO** (Proximal Policy Optimization) is an on-policy reinforcement learning algorithm (Schulman et al., 2017). It efficiently updates its policy within a trust region by penalizing KL divergence or clipping the objective function.

- **SAC** (Soft Actor-Critic) is an off-policy reinforcement learning algorithm (Haarnoja et al., 2018). It learns two Q-functions and utilizes entropy regularization, where the policy is trained to maximize a trade-off between expected return and entropy.

- **TD3** (Twin Delayed Deep Deterministic policy gradient) is an off-policy reinforcement learning algorithm (Fujimoto et al., 2018). It learns two Q-functions and uses the smaller of the two Q-values to form the targets in the loss functions. Additionally, TD3 adds noise to the target action for exploration.

We used the algorithms implemented in stable-baselines-3 (Raffin et al., 2019). The training procedure for an RL agent consists of multiple episodes, and each episode consists of 200 steps. Across each of the experimental configurations, we executed the experiments with 20 different random seeds and present the mean results accompanied by their respective standard deviations. A detailed explanation of RL algorithms and their associated hyperparameters is presented in appendices B.1 and B.2.

### 3.2 Results on controlling 5-stage OPS

In this section, we present the results obtained for the 5-stage OPS system, which involves stacking 32 pulses. We evaluate all four algorithms in three difficulty modes: easy, medium, and hard. It is important to note that SPGD is a training-free method, as it relies on a fixed policy. Therefore, we only evaluate the testing performance of SPGD. For the RL algorithms (PPO, TD3, and SAC), we assess both the training convergence and the testing performance of the trained policy.

The training curves, depicting the reward per step during training iterations, are shown in Fig. 6(a) for the easy mode, Fig. 6(b) for the medium mode, and Fig. 6(c) for the hard mode. From these plots, we observe that TD3 and SAC exhibit similar performance, which is consistently higher than that of PPO across all three difficulty modes. Notably, in the hard mode of the environment, the convergence speed of the algorithms slows down. Moreover, the final convergence value decreases as the difficulty of the environment increases. For instance, in the easy mode, TD3 converges to a reward value of $-0.05$ within 150,000 steps, while it takes 200,000 steps to converge to a reward value of $-0.15$ in the hard mode.

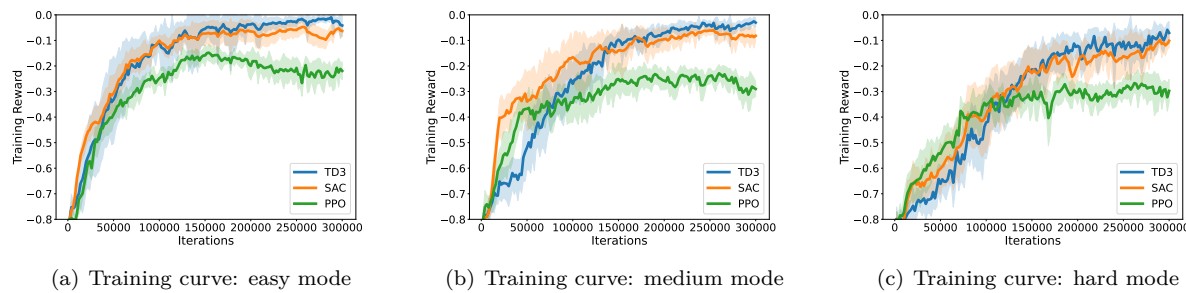

(a) Training curve: easy mode     (b) Training curve: medium mode     (c) Training curve: hard mode

Figure 6: Training curve for SAC, TD3, and PPO on 5-stage OPS environment for (a) easy mode, (b) medium mode, and (c) hard mode. The dashed region shows the area within the standard deviation.

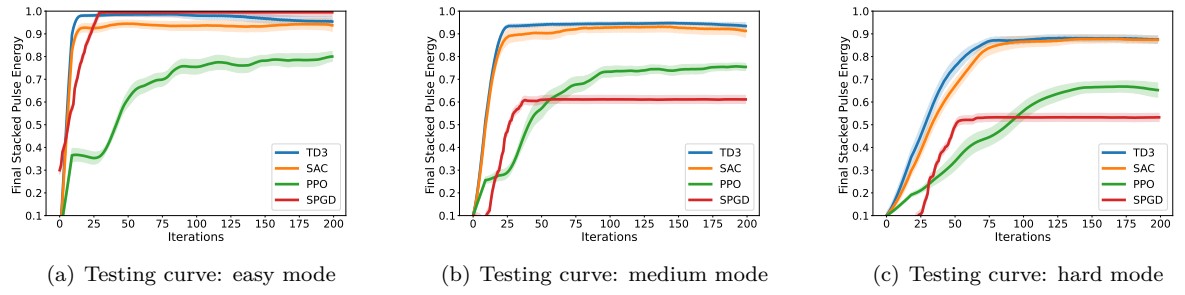

(a) Testing curve: easy mode     (b) Testing curve: medium mode     (c) Testing curve: hard mode

Figure 7: Evaluation of the stacked pulse power $P_N$ (normalized) of different policies in the testing environment for (a) easy mode, (b) medium mode, and (c) hard mode.

Following the training of RL agents, we proceeded to evaluate the performance of the trained policies in the testing environment. The final pulse energy $P_N$ achieved under different iterations is depicted in Fig. 7(a) for the easy mode, Fig. 7(b) for the medium mode, and Fig. 7(c) for the hard mode. We observed that SPGD performs admirably in the easy mode, where the control problem is close to convex. However, its performance deteriorates significantly in the medium and hard modes, which are non-convex control problems. This disparity arises because RL controllers are capable of learning better policies through exploration in non-convex settings. Furthermore, the off-policy RL algorithms (TD3 and SAC) outperform the on-policy RL algorithm (PPO) in our simulation environment. Across all methods, the testing performance in the hard mode is lower than that in the medium mode, despite both being non-convex control problems. This discrepancy can be attributed to the more complex and realistic noise present in the hard mode, which slows down the convergence rate and reduces the final pulse energy in the testing environment.

The final pulse energy $P_N$ is reported for both the training and testing environments, detailing the performance of the trained policy as presented in table 2. It's noteworthy that the training and testing environments for the easy and medium modes exhibit similarity, akin to the classical Atari setup, with performance discrepancies primarily arising from inherent randomness. However, the hard mode introduces distinct noise characteristics between the training and testing environments due to gradual temperature drift. This discrepancy results in a performance gap between the two domains. As depicted, in the easy mode, SPGD significantly outperforms other algorithms, as indicated by Welch's t-test with $p < 0.05$ (Welch, 1947). For the intricate medium and hard modes, TD3 and SAC exhibit superior performance compared to PPO and

SPGD, with statistical significance according to Welch's t-test. Notably, there exist no substantial differences between TD3 and SAC. The fundamental differentiation between SAC, TD3, and PPO lies in their underlying methodologies. SAC and TD3 operate as off-policy approaches, whereas PPO functions as an on-policy method. This can potentially be attributed to PPO's susceptibility to becoming ensnared in local optima due to limited exploration and a tendency to emphasize recent data. On the other hand, SAC and TD3 harness historical experiences, drawn from a replay buffer with a capacity of 10000, thus avoiding undue reliance on only the most recent data. Our observations align harmoniously with insights derived from a benchmark study (Yu et al., 2020). As elucidated in (Yu et al., 2020), SAC achieves successful task resolution across the board, whereas PPO demonstrates adeptness in most tasks while encountering challenges in some cases. The experimental results of different stage OPS environments and discussions can be found in appendices B.3 and B.5.

Table 2: The performance of the implemented algorithms is evaluated based on the Final Pulse Energy $P_N$ (mean $\pm$ standard deviation) in 5-stage OPS environments. The results of the best-performing algorithm on each task, as well as all algorithms that have performances that are not statistically significantly different (Welch's t-test with $p < 0.05$), are highlighted in boldface.

| Mode | Evaluation environment | SPGD | PPO | SAC | TD3 |
|---|---|---|---|---|---|
| easy | training | **0.9919 $\pm$ 0.0121** | 0.8184 $\pm$ 0.0884 | 0.9410 $\pm$ 0.0572 | 0.9552 $\pm$ 0.0483 |
| | testing | **0.9909 $\pm$ 0.0125** | 0.8067 $\pm$ 0.0661 | 0.9403 $\pm$ 0.0531 | 0.9572 $\pm$ 0.0501 |
| medium | training | 0.6155 $\pm$ 0.0271 | 0.7591 $\pm$ 0.0728 | **0.9145 $\pm$ 0.0727** | **0.9353 $\pm$ 0.0451** |
| | testing | 0.6178 $\pm$ 0.0263 | 0.7519 $\pm$ 0.0623 | **0.9098 $\pm$ 0.0838** | **0.9285 $\pm$ 0.0437** |
| hard | training | 0.5321 $\pm$ 0.0352 | 0.6977 $\pm$ 0.0732 | **0.8739 $\pm$ 0.0775** | **0.8724 $\pm$ 0.0680** |
| | testing | 0.5132 $\pm$ 0.0348 | 0.6526 $\pm$ 0.0687 | **0.8371 $\pm$ 0.0804** | **0.8488 $\pm$ 0.0615** |

### 3.3 Run-time Analysis

Our experiments were conducted on an Ubuntu 18.04 system, with an Nvidia RTX 2080 Ti (12 GB) GPU, Intel Core i9-7900x processors, and 64 GB memory. We present the time usage of distinct RL algorithms within a 5-stage OPS environment under hard mode, as delineated in table 3. In the table, "TPS" signifies time usage per step, measured in seconds. The "convergence step" designates the iteration count at which an algorithm attains its peak value, beyond which further iterations yield minimal alterations. During training, the time per step encompasses simulation TPS (denoting OPS simulation time per step) and optimization TPS (reflecting the time spent on gradient descent optimization per step). For instance, TD3 involves a simulation cost of 0.0086s per step and an optimization of 0.0175s per step. With approximately 212018 steps to convergence, the total training duration amounts to about 5456s. Notably, there exists negligible disparity in training times across various algorithms. In the post-training inference phase, the inference TPS represents the time taken for action calculation per step. For instance, TD3's inference cost is 0.0036s per step, with around 58 steps to achieve high pulse energy, leading to a total control time of approximately 0.71 s. As evident, TD3 and SAC distinctly outperform PPO in terms of inference time efficiency, showcasing remarkable superiority. The GPU memory usage of all algorithms is about 1000 MB.

Table 3: Time usage (simulation, optimization, inference time) of the RL algorithms on 5-stage OPS environment with hard mode. The time usage per step (TPS) is quantified in seconds. The algorithm with the optimal time efficiency, as well as algorithms that are not significantly different-performed (Welch's t-test with $p < 0.05$), are highlighted in boldface.

| Time | Simulation | Training | | | Inference | | |
|---|---|---|---|---|---|---|---|
| | simulation TPS | optimization TPS | convergence step | total time | inference TPS | convergence step | total time |
| PPO | 0.0086 | 0.0274 | 142666 $\pm$ 25390 | 5141 $\pm$ 979 | 0.0057 | 78 $\pm$ 10 | 1.13 $\pm$ 0.21 |
| SAC | 0.0086 | 0.0199 | 168844 $\pm$ 24889 | 4833 $\pm$ 740 | 0.0046 | 67 $\pm$ 10 | **0.89 $\pm$ 0.18** |
| TD3 | 0.0086 | 0.0175 | 212018 $\pm$ 28484 | 5456 $\pm$ 804 | 0.0036 | 58 $\pm$ 7 | **0.71 $\pm$ 0.12** |

### 3.4 Comparison of the different settings of OPS environment

In this section, we investigate the impact of different modes (easy, medium, hard) and stage numbers ($N$) in an $N$-stage OPS environment. We evaluate the trained TD3 and SAC policies, as well as SPGD, on different

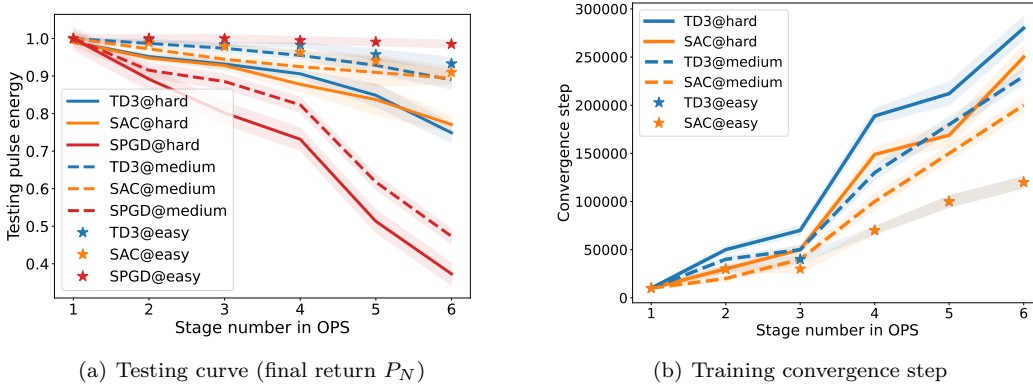

(a) Testing curve (final return $P_N$)

(b) Training convergence step

Figure 8: (a) Final return $P_N$ of different stage OPS on testing environment controlled with TD3 or SAC. (b) Convergence steps for the training of TD3 and SAC on different stage OPS environments. The dashed region shows the area within the standard deviation.

testing environments with varying stage numbers. Figure 8(a) illustrates the final return $P_N$ in relation to the stage number $N$ in the $N$-stage OPS environment, comparing the performance of different algorithms across different modes. From the figure, we draw the following conclusions: (1) For the easy mode, SPGD outperforms SAC and TD3, and all methods achieve almost-optimal performance regardless of the stage number $N$. (2) Across both the medium and hard modes, the performance of off-policy RL methods (SAC and TD3) is comparable, and RL methods are better than SPGD. (3) As the stage number increases in the hard and medium modes, the performance of SPGD drops significantly. Moreover, for $N \geq 4$, RL methods (SAC and TD3) outperform SPGD significantly (under Welch's t-test). Figure 8(b) illustrates the training convergence steps for different stage OPS. It can be observed that as the stage number increases, the number of steps required for training convergence also increases significantly.

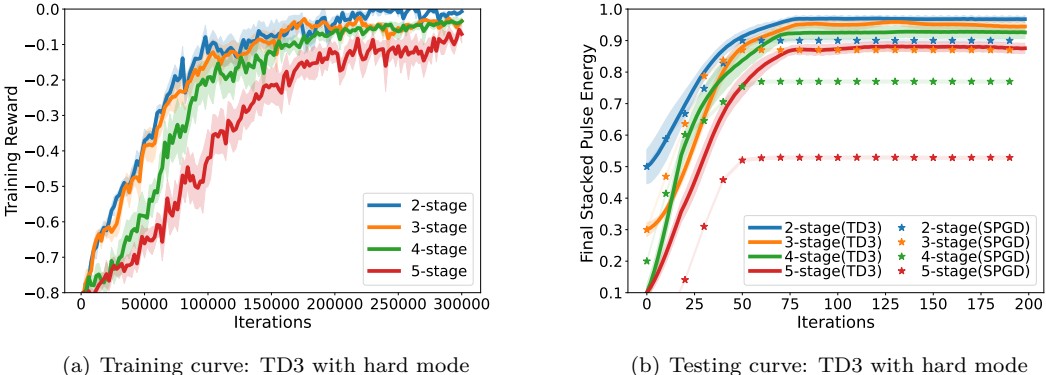

(a) Training curve: TD3 with hard mode

(b) Testing curve: TD3 with hard mode

Figure 9: Comparison of the results on hard mode $N$-stage OPS environment with TD3 algorithms. (a) shows the training curve; (b) shows the evaluation of TD3 and SPGD in the testing environment. The dashed region shows the area within the standard deviation.

To provide a clearer illustration of the impact of stage number $N$ in the OPS environment, we present the training and testing curves for TD3 and SPGD on the hard mode. Figure 9(a) displays the training curve of TD3 on different $N$-stage OPS systems. It can be observed that as the stage number increases, the training convergence becomes slower. Figure 9(b) showcases the testing curve of SPGD and TD3 on different $N$-stage OPS systems. From the figure, we can draw the following observations: (1) With an increase in stage number, the final return $P_N$ becomes smaller for both TD3 and SPGD, indicating a decrease in performance as the system becomes more complex. (2) TD3 consistently outperforms SPGD for stage numbers $N \geq 4$.

This suggests that RL methods like TD3 are more effective in handling the control challenges posed by OPS systems with a larger number of stages.

### 3.5 Transferring trained policy between different modes

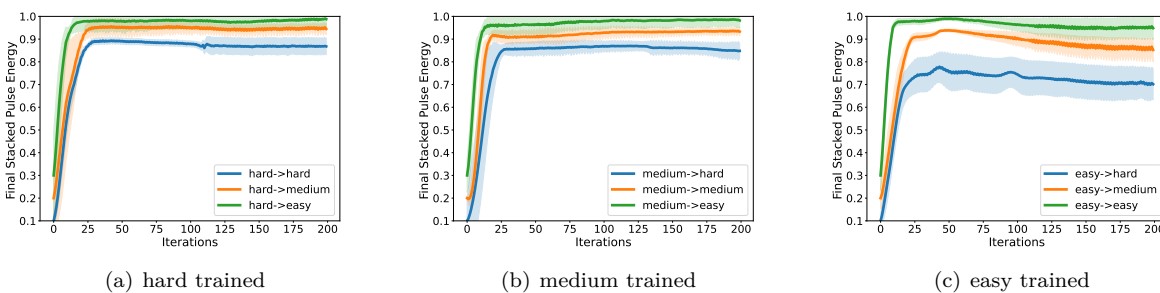

(a) hard trained        (b) medium trained        (c) easy trained

Figure 10: Demonstration of the transfer performance of the trained policy on (a) hard mode training environment; (b) medium mode training environment; (c) easy mode training environment. The dashed region shows the area within the standard deviation.

The major difference between the simulation and real-world environments is the different noise levels. In order to investigate the transferability of trained policies between different noise levels in the OPS environment, we conducted a simulated experiment. Our simulation environment incorporates different noise levels depending on the difficulty mode: "easy", "medium", and "hard". We trained policies on the "hard" mode environment and tested their performance on "hard", "medium", and "easy" mode environments. The transfer results are presented in Fig. 10(a). As can be observed, the trained policy can be successfully transferred to both "medium" and "easy" mode environments, achieving high performance in terms of pulse energy. Figure 10(b) and Figure 10(c) depict the transfer results of policies trained on the "medium" and "easy" mode environments, respectively. It can be seen that when policies trained on easier environments are transferred to a harder environment, their performance drops. On the contrary, policies trained in harder environments can be effectively applied to easier environments. Based on these findings, training policies in harder simulation environments that introduce more noise and uncertainty can be more useful. This approach allows us to explore and develop fast and robust control algorithms that can then be deployed on real-world physical systems.

## 4 Discussion

### 4.1 Real-world environment and simulation

Deploying RL algorithms in real-world optics systems poses several challenges, including the need for signal conversion, time delays, and manual tuning of optical devices. In our simulation system, we have the advantage of faster control steps and simplified initial alignment. In real-world optics systems, the optical signal needs to be converted to an electrical analog signal using a photo-detector (PD), which is then further converted to a digital signal using an analog-to-digital converter (ADC). Furthermore, the delay line device or controller introduces supplementary time overhead to the control step, typically spanning from 0.1 to 1 second. Nonetheless, within our simulation system, both simulation and control steps can be performed in less than 0.01 seconds, as demonstrated in table 3. This facilitates expedited training and evaluation processes. Furthermore, in real-world OPS systems, manual tuning of the optical devices is required when the optical beams are misaligned, which can be a time-consuming process taking several hours or even days. In contrast, in our simulation system, we can easily reset the environment to achieve the initial alignment, simplifying the setup and reducing the time required for system preparation.

Previous endeavors to directly train RL algorithms to real-world OPS systems have encountered obstacles, including slow training in a real environment, and unstable and non-optimal convergence of RL algorithms. Consequently, the sim2real approach, which involves training and evaluating RL algorithms in simulation environments before deploying them to real-world systems, has garnered considerable interest. Our objective

is to conduct comprehensive research on RL algorithms within the simulation environment and subsequently leverage the sim2real approach to transition these algorithms into real-world applications.

The validity of our simulation is partially confirmed in (Tünnermann & Shirakawa, 2019; Yang et al., 2020). The experiments detailed in (Tünnermann & Shirakawa, 2019) align with the 1-stage OPS environment described within our paper. In their study, the authors conducted both simulations and real experiments to assess RL algorithms and demonstrated their usefulness in the context of their paper. They also emphasized the value of using simulations for training due to the time and effort required for real-world RL training. Although our OPS simulation and experimental settings are more intricate than those in (Tünnermann & Shirakawa, 2019), the underlying physics remains consistent, bolstering the reliability and accuracy of our simulation environment.

### 4.2 RL controllers and different simulation modes

The experimental results demonstrate that off-policy RL algorithms (TD3 and SAC) outperform traditional SPGD controllers in larger $N$-stage OPS systems, particularly in the challenging hard mode. In the simulation, the easy mode corresponds to the traditional control approach in which experts manually fine-tune the OPS system to achieve a state close to the global optimum (representing the real-world "easy" mode) before employing SPGD for control. However, the future of optical control lies in automation. The hard mode in the simulation reflects a more realistic scenario where direct control of the OPS system is performed without initial expert tuning. In this context, RL controllers exhibit significant promise for optical systems. This motivates our focus on developing OPS simulation environments, emphasizing the need for fast training and noise-robust RL algorithms capable of handling non-stationary noise and non-convex control objectives. Additionally, exploring the nonconvex and periodic nature of OPS objectives holds potential benefits for real-world RL applications, incorporating valuable structural information into the control tasks in optics.

### 4.3 Limitations

In our simulation, we model vibration-induced fast noise as Gaussian noise. However, it's important to acknowledge that vibrations may not consistently conform to a Gaussian distribution. The characteristics and features of noise generated by vibrations can vary based on a range of factors. For instance, if vibration properties exhibit non-symmetry or involve non-linear effects, the resulting noise profile might deviate from Gaussian attributes. Guided by the Central Limit Theorem, we consider Gaussian noise to be a suitable approximation for representing fast noise in our simulation. As a result, the principal disparity between real-world experiments and simulations often originates from divergent noise properties. In forthcoming research, we aim to delve into sim2real transfer and formulate a more realistic gap, taking into account these nuanced noise characteristics.

## 5    Conclusion

In this paper, we present OPS, an open-source simulator for controlling pulse stacking systems using RL algorithms. To the best of our knowledge, this is the first publicly available RL environment specifically tailored for optical control problems. We conducted evaluations of SAC, TD3, and PPO within our proposed simulation environment. The experimental results clearly demonstrate that off-policy RL methods outperform traditional SPGD-PID controllers by a substantial margin, especially in challenging environments. By offering an optical control simulation environment and providing RL benchmarks, our aim is to encourage the exploration and application of RL techniques in optical control tasks, as well as to facilitate further advancements of RL controllers in the field of natural sciences.

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

# A Related Works

## A.1 Optical Pulse Stacking

High-energy lasers find extensive application in scientific research, medical procedures, and various industries (Fermann & Hartl, 2013). Nonetheless, the pulse energy of individual lasers often falls short for applications such as large-scale industrial material processing (Ready, 1997). This limitation arises due to the non-linearities and losses inherent in laser resonators (Siegman, 1986). Overcoming this challenge necessitates innovative approaches to achieve higher energies without being constrained by the limitations of a single laser pulse. Optical (coherent) pulse stacking has emerged as a solution to this predicament (Tünnermann & Shirakawa, 2017), offering a promising avenue for scaling pulse energy. The nonlinearity and periodicity inherent to light interference, underlie not only OPS but also other optical control challenges. These techniques are integral to precision measurements, industrial manufacturing, and scientific investigations. Our simulation, which mirrors these optical control complexities, represents a significant and representative environment for optical control experimentation.

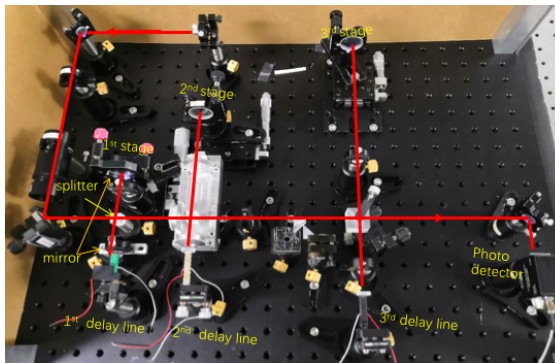

Figure 11: Real-world optical pulse stacking system. A controller adjusts time delay values to achieve maximum pulse energy.

The actual OPS system is depicted in Fig. 11. A controller is responsible for determining the value of each time delay $\tau$ by measuring the final stacked pulse using a photodetector. Subsequently, these time delay values are transmitted to each delay line device in order to adjust the positions of the pulses. It should be noted that in Fig. 11, the controller is connected to the electric signal line of the 1st, 2nd, and 3rd delay line devices located at the bottom. Conducting real-world OPS control experiments can be both costly and time-consuming due to the complexities involved in the process. To visually illustrate the stacking procedure for combining two pulses, please refer to Supplemental Video 1 [2].

## A.2 Reinforcement Learning Benchmarks

In tandem with the remarkable achievements of Deep Reinforcement Learning (RL), a plethora of Reinforcement Learning Benchmarks have been explored. These include well-known platforms like the Atari Arcade Learning Environment (Bellemare et al., 2013), the OpenAI gym (Brockman et al., 2016), and the DeepMind Control (DMC) Suite (Tassa et al., 2018). Beyond these general benchmarks, the advancement of Reinforcement Learning across diverse applications has spurred the development of specialized benchmarks. Noteworthy examples include Rllab (Duan et al., 2016), tailored for continuous control challenges, and Meta-World (Yu et al., 2020), serving as a benchmark for Meta Reinforcement Learning. SafetyGym (Ray et al., 2019) evaluates the ability of RL agents to accomplish tasks while adhering to safety constraints, while URLB (Laskin et al., 2021) assesses the performance of Unsupervised Reinforcement Learning. While these existing benchmarks aptly cater to control problems in the fields of Robotics and Machine Learning, a notable gap exists in the availability of a benchmark and repository of user-friendly baseline algorithms for scientific control problems or optical control tasks. This significant gap is our primary motivation for propelling progress in the realm of RL benchmarks for optical control problems through OPS.

---

[2]https://github.com/Walleclipse/Reinforcement-Learning-Pulse-Stacking/blob/main/demo/Video1.gif

# B  Additional details of Experiments

## B.1  RL algorithms

Reinforcement Learning (RL) is a field of machine learning focused on how intelligent agents should make decisions in an environment to maximize cumulative rewards (Sutton & Barto, 2018). In our OPS environment, RL algorithms learn a policy that aims to maximize the output pulse power by controlling time delay, as detailed in section 2.3. For this study, we conducted a thorough evaluation of three RL algorithms: Proximal Policy Optimization (PPO), Twin Delayed Deep Deterministic Policy Gradients (TD3), and Soft Actor-Critic (SAC). We briefly illustrate each algorithm as follows.

- **Proximal Policy Optimization (PPO)** is a popular policy gradient algorithm used in reinforcement learning (Schulman et al., 2017). PPO belongs to the class of on-policy algorithms, meaning it optimizes the policy by collecting data through interactions with the environment in real-time. The key idea behind PPO is to ensure that the policy updates are not too large, preventing significant policy changes that might lead to instability or catastrophic forgetting. Specifically, PPO uses a clipped surrogate objective function to update the policy. The objective aims to maximize the expected reward while also introducing a constraint that prevents the policy update from deviating too far from the current policy.

- **Soft Actor-Critic (SAC)** is one of the commonly used deep reinforcement learning algorithms introduced by (Haarnoja et al., 2018). SAC is an off-policy algorithm, which means it can learn from past experiences (replay buffer) and does not rely on the most recent data like on-policy algorithms. Off-policy learning enables SAC to efficiently use data and improve sample efficiency. One of the key features of SAC is the emphasis on entropy maximization. The policy is not only optimized to maximize the expected reward but also to maximize the entropy of the policy. By maximizing the entropy, SAC encourages exploration and avoids premature convergence to suboptimal policies. Another key feature is the soft value function. SAC uses soft value functions instead of the typical Q-functions used in other algorithms. These soft Q-functions estimate the expected return, but they are "softened" by adding an entropy term.

- **Twin Delayed Deep Deterministic policy gradient (TD3)** is a powerful deep reinforcement learning algorithm designed for solving tasks with continuous action spaces (Fujimoto et al., 2018). Similar to SAC, TD3 is an off-policy algorithm, which means it could efficiently use data and improve sample efficiency. TD3 follows the actor-critic framework. It consists of two neural networks: an actor-network and two critic networks. The actor-network learns the policy, mapping states to continuous actions, while the critic networks estimate the Q-values (expected return) for different state-action pairs. The actor network learns a deterministic policy, allowing for easier optimization in continuous action spaces. TD3 employs two critic networks to reduce overestimation biases commonly encountered in Q-learning algorithms.

For further information on RL algorithms, readers can refer to the comprehensive survey paper by (Wang et al., 2020).

## B.2  Experimental setting

To optimize the performance of RL algorithms, we conducted a grid search to identify optimal hyperparameters. Each set of hyperparameters was thoroughly evaluated using 5 different random seeds. The search process involved training on the 5-stage OPS environment with medium difficulty. Detailed information regarding the hyperparameter ranges and the selected values for TD3, SAC, and PPO can be found in tables 4 to 6. While we didn't explicitly mention the specific hyperparameter values except in tables 4 to 6, we maintained the default settings provided by the stable baseline3 library.

Model size stands as a pivotal hyperparameter, bearing significance not only on evaluation performance but also exerting influence on training duration. Table 7 presents a juxtaposition of diverse model sizes for the policy network within the TD3 algorithm and the 5-stage OPS environment. In this context, [64,64] signifies a fully connected network boasting two layers, each with a hidden dimension of 64. Conversely, [256,256] represents a fully connected network encompassing three layers, with a hidden dimension of 256. Evidently, the network architecture of [256,256] emerges as the most proficient, excelling both in the assessment of

Table 4: TD3: ranges used during the hyperparameter search and the final selected values.

| Hyperparameter | Range | Best-selected |
|---|---|---|
| Size of the replay buffer | {1000,10000,100000} | 10000 |
| Step of collect transition before training | {100, 1000, 10000} | 1000 |
| Unroll Length/$n$-step | {1,10, 100} | 100 |
| Training epochs per update | {1,10, 100} | 100 |
| Discount factor ($\gamma$) | {0.98, 0.99, 0.999} | 0.98 |
| Noise type | {'normal', 'ornstein-uhlenbeck', None} | 'normal' |
| Noise standard value | {0.1, 0.3, 0.5, 0.7, 0.9} | 0.7 |
| Learning rate | {0.0001, 0.0003, 0.001,0.003,0.01} | 0.0003 |
| Policy network hidden layer | {2, 3} | 2 |
| Policy network hidden dimension | {64, 128, 256} | 256 |
| Optimizer | Adam | Adam |

Table 5: SAC: ranges used during the hyperparameter search and the final selected values.

| Hyperparameter | Range | Best-selected |
|---|---|---|
| Size of the replay buffer | {1000,10000,100000} | 10000 |
| Step of collect transition before training | {100, 1000, 10000} | 1000 |
| Unroll Length/$n$-step | {1,10, 100} | 1 |
| Training epochs per update | {1,10, 100} | 1 |
| Discount factor ($\gamma$) | {0.98, 0.99, 0.999} | 0.99 |
| Generalized State Dependent Exploration (gSDE) | {True, False} | True |
| Soft update coefficient for "Polyak update" ($\tau$) | {0.002,0.005, 0.01, 0.02} | 0.002 |
| Learning rate | {0.0001, 0.0003, 0.001,0.003,0.01} | 0.0003 |
| Policy network hidden layer | { 2, 3} | 2 |
| Policy network hidden dimension | {64, 128, 256} | 256 |
| Optimizer | Adam | Adam |

Table 6: PPO: ranges used during the hyperparameter search and the final selected values.

| Hyperparameter | Range | Best-selected |
|---|---|---|
| Unroll Length/$n$-step | {128,256,512,1024,2048} | 512 |
| Training epochs per update | {1,5,10} | 10 |
| Clipping range | {0.1,0.2,0.4} | 0.2 |
| Discount factor ($\gamma$) | {0.98, 0.99, 0.999} | 0.99 |
| Entropy Coefficient | {0, 0.001, 0.01, 0.1} | 0.001 |
| GAE ($\lambda$) | {0.90, 0.95, 0.98, 0.99} | 0.95 |
| Value function coefficient | {0.1,0.3,0.5,0.7,0.9} | 0.5 |
| Learning rate | {0.0001, 0.0003, 0.001,0.003,0.01} | 0.0003 |
| Gradient norm clipping | {0.1, 0.5, 1.0, 5.0} | 0.5 |
| Policy network hidden layer | {2, 3} | 2 |
| Policy network hidden dimension | {64, 128, 256} | 256 |
| Optimizer | Adam | Adam |

stacked pulse energy (Evaluation performance) and time efficiency (Training Time Usage). Consequently, we opt for the [256,256] architecture as our policy network configuration.

Table 7: Comparative evaluation of various models within the TD3 algorithm and 5-stage OPS environment, with emphasis on performance, memory usage, and time costs.

| Model size | Evaluation performance | Memory Usage | Training Time Usage |
|---|---|---|---|
| $[64, 64]$ | $0.8931 \pm 0.0873$ | 933 MB | $\approx 83$ min |
| $[256, 256]$ | $0.9285 \pm 0.0437$ | 989 MB | $\approx 90$ min |
| $[256, 256, 256]$ | $0.9243 \pm 0.0399$ | 1030 MB | $\approx 117$ min |

## B.3 Results on controlling 4-stage and 6-stage OPS environments

We present the training curves (training reward vs. iterations) and testing curves (final pulse energy $P_N$ vs. testing iterations) for the 4-stage OPS environment in Fig. 12 and for the 6-stage OPS environment in Fig. 13. From the figures, it is evident that TD3 and SAC outperform PPO in terms of performance. Moreover, comparing Fig. 12 (4-stage) to Fig. 13 (6-stage), it can be observed that as the stage number increases, the training convergence becomes slower and the final return $P_N$ becomes smaller, particularly in the medium and hard difficulty modes.

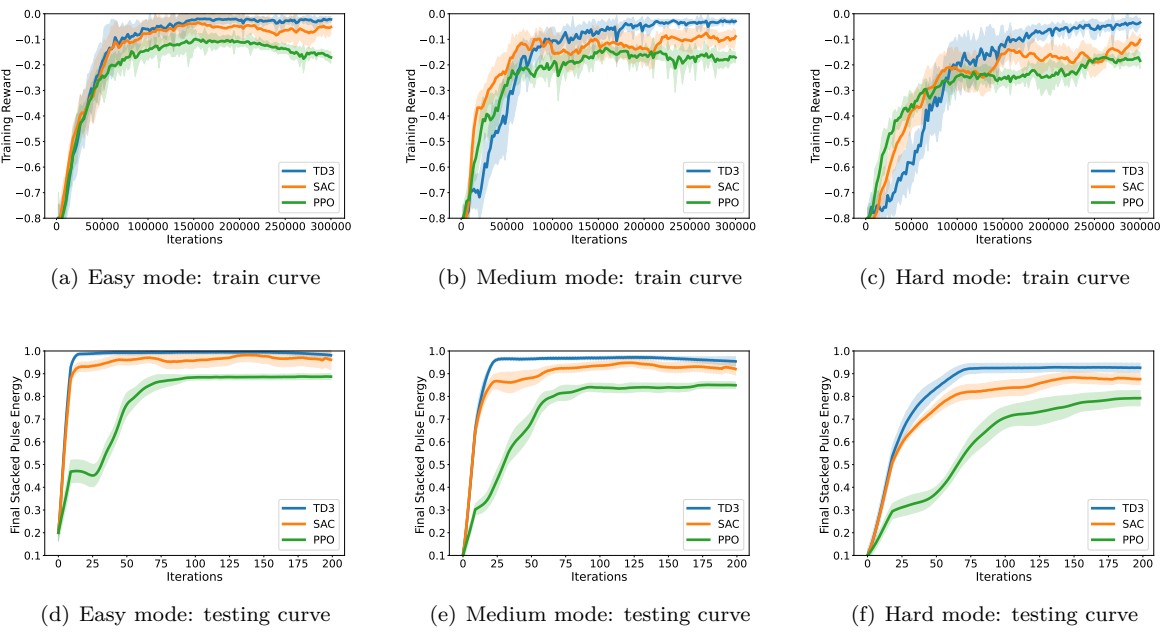

(a) Easy mode: train curve     (b) Medium mode: train curve     (c) Hard mode: train curve

(d) Easy mode: testing curve     (e) Medium mode: testing curve     (f) Hard mode: testing curve

Figure 12: 4-stage OPS experiments. Training reward was plotted for (a) easy mode, (b) medium mode, and (c) hard mode. Evaluation of the stacked pulse power $P_4$ (normalized) of the testing environment was plotted for (d) easy mode, (e) medium mode, and (f) hard mode.

## B.4 Rendering the controlling results on OPS environment

Figure 14 depicts the pulse trains on a 5-stage hard mode OPS system controlled by TD3, starting from a random initial state. It is evident from the figure that the TD3 algorithm is capable of attaining a maximum power within 40 iterations. For a more comprehensive visualization, please refer to supplemental video 2 [3].

---

[3] https://github.com/Walleclipse/Reinforcement-Learning-Pulse-Stacking/blob/main/demo/Video2.gif

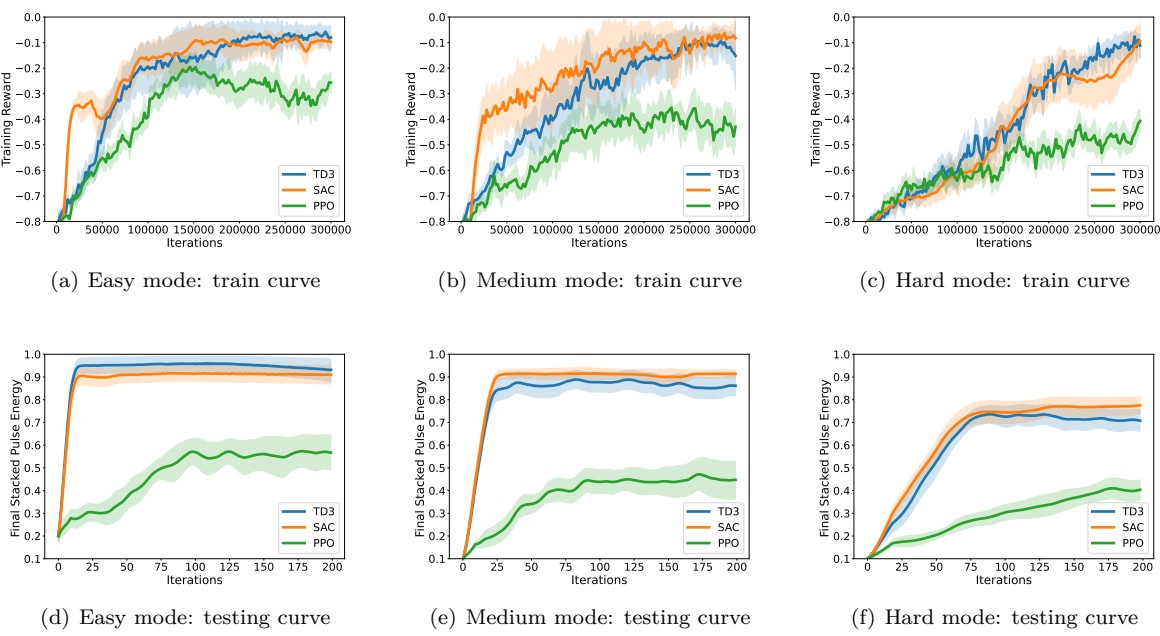

Figure 13: 6-stage OPS experiments. Training reward was plotted for (a) easy mode, (b) medium mode, and (c) hard mode. Evaluation of the stacked pulse power $P_6$ (normalized) of the testing environment was plotted for (d) easy mode, (e) medium mode, and (f) hard mode.

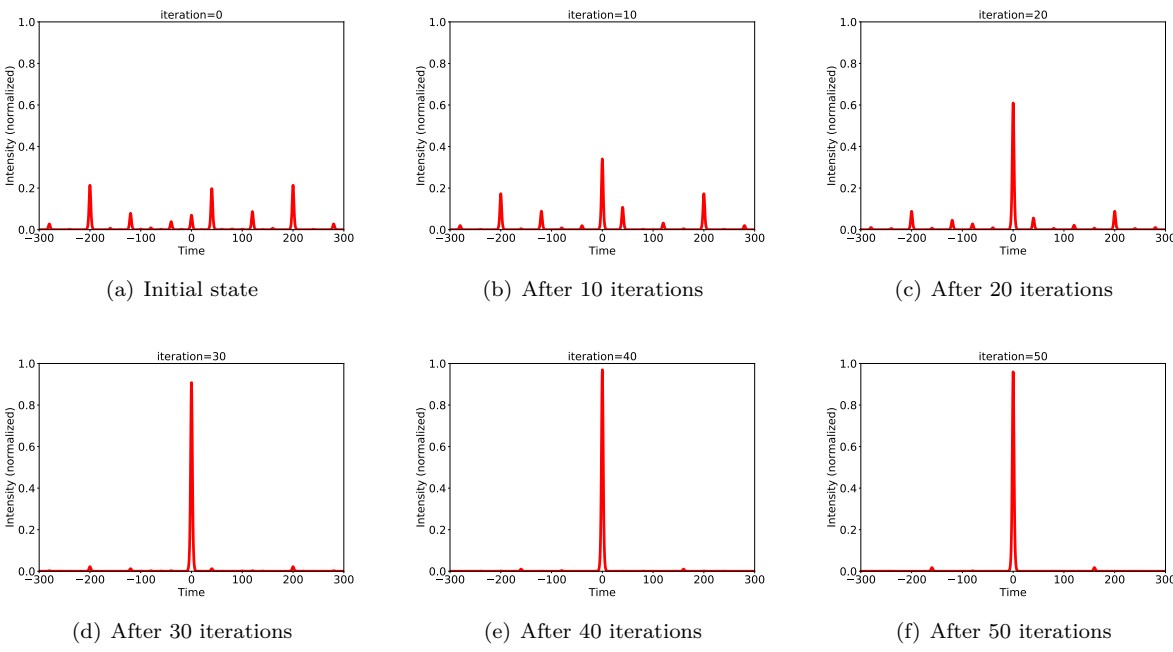

Figure 14: Demonstration of the controlling 5-stage OPS hard mode testing environment by TD3 algorithm after training. (a): initial state of pulses; (b) pulse state after 10 control iterations; (c) pulse state after 20 control iterations; (d) pulse state after 30 control iterations; (e) pulse state after 40 control iterations; (e) pulse state after 50 control iterations.

### B.5 Discussion about experimental results

In our experiments, we observed a significant performance advantage of SAC and TD3 over PPO. The primary distinction between SAC, TD3, and PPO lies in their underlying approach - SAC and TD3 are off-policy methods, while PPO is an on-policy method. The fundamental characteristic of on-policy algorithms is that both the data used for learning and policy improvement originate from the same policy being updated. This implies that the agent relies heavily on recent experiences to adjust its policy (Sutton & Barto, 2018). In the context of our OPS environment, where numerous local optimal points exist (as depicted in Figure 2), PPO can become trapped in local optima due to its limited exploration and tendency to focus on recent data, limiting its ability to discover globally optimal policies. Conversely, SAC and TD3 can leverage historical experiences (e.g., from a replay buffer of size 10000) and are not constrained to relying solely on the most recent data. This typically results in more efficient data utilization and potentially better convergence towards optimal solutions compared to on-policy methods. Our observations align with findings from a benchmark study (Yu et al., 2020). As demonstrated in Figures 6 and 8 of (Yu et al., 2020), SAC successfully solves all tasks presented in the paper, while PPO manages to solve most tasks but falls short in certain cases. We acknowledge that PPO might exhibit better performance in specific scenarios. However, considering the results presented in both our paper and (Yu et al., 2020), it appears that SAC may serve as a more promising starting point for addressing control problems using RL algorithms.

### B.6 Feedback delay

In control systems, the presence of feedback loops invariably introduces delays due to finite sensing speeds, signal processing, computation of control inputs, and actuation. Specifically, let $\tau_{\text{real}}(t)$ denote the time delay (or state) of a real OPS system at time step $t$. Consider that activities such as photo-detection, data acquisition, analog-to-digital conversion, and action computation require a time interval $\delta$. Consequently, a feedback delay of $\delta$ emerges within our OPS control system. When we execute action $a_t = a_t(\tau_{\text{real}}(t))$, the system's time delay (state) might deviate from $\tau_{\text{real}}(t)$ to $\tau_{\text{real}}(t) + D(\delta)$ due to the feedback delay. Here, $D(\delta)$ accounts for an additive time-delay value attributed to the feedback delay $D(\cdot)$. The the next step's time-delay can be expressed as:

$$\tau_{\text{real}}(t + 1) = \tau_{\text{real}}(t) + a_t + D(\delta). \tag{6}$$

In our simulation, we replicate the feedback delay (or hardware-dependent delay) by injecting free-running noise, including elements like slow noise and fast noise, into the delay lines. This process is elucidated in (4), which bears a resemblance to the representation in (6). This noise effectively emulates the sequence of events seen in real-world experiments, encompassing photo-detection, data preprocessing, and the computation of actions.

The proposed OPS environment offers versatile noise levels, allowing for the configuration of distinct feedback delays and associated system free-running noise. In this context, we present an exploration involving the training of RL algorithms within a low feedback delay system, followed by the transfer of the learned policy to a system characterized by a higher feedback delay. The training noise within the 5-stage hard-mode OPS system aligns with the feedback delay $\delta$ and noise $e_t = D(\delta)$ , exemplified by $\delta = 10$ ms. Subsequently, the policy trained under these conditions is evaluated within systems featuring elevated feedback levels $k\delta$ and noise $e_{new} = D(k \cdot \delta)$, where $k$ takes on values such as 1, 20, 40, 60, 80, and 100. As depicted in Fig. 15, it is evident that, regardless of the specific feedback delay, the trained policy consistently attains convergence within 70 steps, thereby ensuring a cumulative pulse power exceeding 0.7 (a.u.). However, it is noteworthy that an increase in feedback delay corresponds to a notable decline in performance, particularly concerning pulse energy.

## C  Potential Impact and future work

Our simulation environment offers significant benefits in tackling challenging and realistic reinforcement learning problems. Real-world reinforcement learning problems are often highly challenging due to factors such as high-dimensionality of control, noisy behaviors, and distribution shift (Dulac-Arnold et al., 2019; Agarwal et al., 2021; Du et al., 2019). By selecting a large $N$-stage number with the hard mode in our

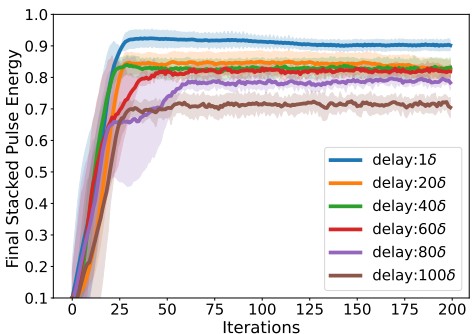

Figure 15: Subsequent to the training of a TD3 policy within a low feedback delay OPS system characterized by $\delta$, we then evaluate the policy's performance within systems characterized by higher feedback delays $k \cdot \delta$.

simulation environment, we can create high-dimensional and difficult control scenarios. The hard mode of the OPS environment exhibits a distinct noise distribution in the testing environment compared to the training environment, which mirrors the challenges encountered in real-world reinforcement learning problems.

In our simulation, we have access to the objective function of the OPS (ignoring noise), which provides valuable structural information and physical constraints. This enables us to explore additional information about the OPS function and incorporate it into RL algorithms. Rather than focusing on generic nonconvex problems, many real-world scenarios involve specific nonconvex control problems with known objective functions or physical constraints (Miryoosefi et al., 2019). Exploring the nonconvex and periodic nature of the OPS objective can greatly benefit real-world RL problems that encompass structural information.

Similar to our OPS control system, optical control problems in general are influenced by the nonlinearity and periodicity of light interactions. This includes applications such as coherent optical interference (Wetzstein et al., 2020) and linear optical sampling (Dorrer et al., 2003), which find utility in precise measurement, industrial manufacturing, and scientific research. We consider our simulation environment to be an important and representative optical control environment. Furthermore, RL methods have the potential to drive advancements in optical laser technologies and the next generation of scientific control technologies (Genty et al., 2020).

