# OpenReview forum: "An Optical Control Environment for  Benchmarking Reinforcement Learning Algorithms"
_TMLR — Accepted by TMLR_

### Review · Reviewer_C8oN · 2023-07-22

**Summary Of Contributions:**

# Summary of Contributions

The Paper _An Optical Control Environment for Benchmarking Reinforcement Learning Algorithms_ introduces a new environment for reinforcement learning called _Optical Pulse Stacking_ (OPS). OPS is a scalable simulator of optical control systems and will be released as open-source code. Currently, there are no open-source implementations of optical control simulated reinforcement learning environments, and OPS could help advance optical control through automation using reinforcement learning algorithms.

**Audience:**

Yes

**Broader Impact Concerns:**

# Broader Impact and Ethical Concerns

No ethical concerns.

**Claims And Evidence:**

Yes

**Requested Changes:**

# Critical Changes

See the _Main Argument_ above.

I would be happy to increase my recommendation to an _accept_ if more runs were used as outlined in the final paragraph of the _Main Argument_ section above.

# Changes which would simply strengthen the work

See _Small Things_ above.

**Strengths And Weaknesses:**

# Main Argument

**Recommendation**: ~Reject~ Accept

Overall, I believe that the OPS environment could be beneficial to many researchers in the areas of reinforcement learning and optical control and could be a significant contribution to these fields. Unfortunately, the experiments provided in the paper do not fully substantiate the usefulness of the OPS environment.

My main concern is that a number of the claims made in the _Experiments_ section and appendix are not fully supported by statistically significant data. If more random seeds were used, then the claims could potentially be supported by statistically significant data.

In section 3.2, multiple claims are made which are not supported by statistically significant data. The caption for the relevant figure, Figure 6(a), states that the dashed (shaded) regions are standard deviations (standard errors). With normality assumptions, a 95% confidence interval would be roughly double the size of this region, and the confidence intervals would likely overlap for all algorithms in these plots. Likely, this normality assumption is too strong and the confidence intervals would be even larger than this. Furthermore, when considering confidence intervals for a specific algorithm across plots, the confidence intervals would also likely overlap (for example, considering the confidence intervals of PPO in both Figure 6(a) and Figure 6(c)). Because of this, a number of claims are not supported by this data:

1. TD3 and SAC perform better than PPO
2. The final, converged-to value decreases as the difficulty increases

Furthermore, in section 3.2 the paper makes the claim that in the hard environment, the convergence speed slows down. I am not fully convinced of this statement based on Figure 6. For example, TD3 and SAC both seem to converge around 150,000 steps for the medium and hard modes. The claim could likely be supported by increasing the statistical significance of the data shown (by using more runs) or by using a more measured claim. For example "compared to the easy mode environment, the convergence speed is slower in the hard mode environment for all algorithms". How was convergence speed measured here? Perhaps my evaluation of convergence is incorrect.

Similar claims to those above are also made in Appendix B, where similar experiments are run for the 4-stage and 6-stage OPS environment. Again, the data does not appear to be statistically significant and support the claims:

1. TD3 and SAC outperform PPO
2. As the stage number increases, the final return P_N becomes smaller

In Section 3.3, the claim that the rate of convergence decreases as the stage number of the OPS environment increases does not appear to be supported by statistically significant data in Figure 9(a). I could not find in the paper an explanation of the lines and shaded regions in Figure 9(a), but I am assuming the lines are the mean performance and the shaded regions are standard errors across the 10 runs. Can the authors confirm if this is the case? If this is the case, then as above, the relevant confidence intervals would be significantly overlapping.

In Section 3.4, how many runs are used for these transfer experiments? I am assuming only a single run was used (since no shaded regions are plotted). If this is the case, why was only one run used? Is one run enough to verify the hypotheses made in this section? Perhaps there is some part of the environment I am misunderstanding which warrants the use of a single run here, or perhaps more runs actually were used and my assumption is wrong. Or maybe this sections is simply for a proof-of-concept, showing that it is _possible_ to train on one version of the environment and do fairly well on the other versions of the environment. If this is the case, it might be good to clarify.

Some experiments do not provide sufficient experimental details. For SAC, was the entropy scale hyperparameter swept? If so, what values were swept, and which value was selected? If not, what value was used? Why were episodes cut off at 200 steps? Episode cutoffs are known to affect learning dynamics, with longer episodes typically being more difficult [1]. How was convergence measured?

Another note is that these reinforcement learning algorithms may not be fully tuned. The hyperparameters were chosen using 3 random seeds. It is well known that these algorithms exhibit large variation across random seeds [2]. It could be the case that due to this randomness, incorrect hyperparameters were chosen. If the results in the paper are to serve as a benchmark of performance against which future researchers can compare their algorithms, then likely using more runs when tuning these hyperparameters would be useful.

Finally, is hard mode actually more difficult than medium mode of the OPS environment? In many of the figures, the performance of each algorithm on medium mode is comparable to its performance on hard mode (and typically not statistically significant). Then in Figure 10 (a,b), transferring from training on medium mode to testing on hard mode results in similar final performance as training and testing on hard mode. Can the difficulty of hard mode be adjusted by adjusting the level of noise? How similar to a real optical stacking controller is hard mode (this might not even be possible to check)? Maybe this question would be answered by having more experimental runs, which would better show the difference in performance of each algorithm on medium and hard mode.

In summary, OPS could be a significant contribution to the fields of reinforcement learning and optical control. Unfortunately, the utility of the environment comes into question due to the lack of statistically significant data to support the claims made by the paper. I would be happy to change my recommendation to _accept_ if more runs were used such that the **experimental claims** which I noted above were supported by statistically significant data. Clearly, upon publication, the results in this paper will be used to benchmark new algorithms against, and so it is crucial that the results presented here be statistically significant and of high confidence.

# Small Things

**These did not affect the scoring of the paper**

- Algorithm references are not in parentheses in Section 3.1
- “We implement the RL algorithms using stable-baseline-3“ → “We used the algorithms implemented in stable-baselines-3”
- Y axis label in Figure 8(b) is cut off
- Section 4.1: “We In real-world optics systems...” → “In real-world optics systems...”

# References

[1] Andrew Patterson, Samuel Neumann, Martha White, Adam White. Empirical Design in Reinforcement Learning. 2023.

[2] Peter Henderson, Riashat Islam, Philip Bachman, Joelle Pineau, Doina Precup, David Meger. Deep Reinforcement Learning that Matters. 2018.

---

> ### Author Response · Authors · 2023-08-07
> **Response to Reviewer C8oN**
>
> 1) My main concern is that a number of the claims made in the Experiments section and appendix are not fully supported by statistically significant data. If more random seeds were used, then the claims could potentially be supported by statistically significant data.
> A: Thank you for your feedback.
> To enhance the robustness and reliability of our results, we extended our experimentation to include a more substantial number of random seeds. Specifically, we conducted experiments using 20 different random seeds. This deliberate approach exceeds the prevailing practice in related literature. For instance, we note that [1] utilized 5 random seeds, while [2] employed 10 random seeds for their experiments.
> Moreover, we have incorporated Welch’s t-test [3] as a statistical validation method. It allows us to rigorously evaluate the statistical significance of the differences observed between various algorithms.  We have updated the pertinent results in both Section 3 and Appendix B to reflect these changes.
> As a direct outcome of these improvements, we present refined conclusions in terms of the evaluation performance of the stacked pulse power post-training. Specifically, for the medium and hard modes, SAC and TD3 stand out significantly against SPGD and PPO. Notably, TD3 and SAC showcase comparable performance, eliminating any significant performance gap. This consistent trend holds across the 4,5,6-stage scenarios.
> In terms of training performance, as depicted in Figure 6, we observe SAC and TD3 outperforming SPGD and PPO in the medium mode (Fig 6(b)). However, in the hard mode, the statistical significance is not as clear. Similarly, for the metric of training convergence steps, we do not find statistically significant differences among TD3, SAC, and PPO.
> It's important to note that our focus leans more toward evaluation performance rather than training curves. The variability in noise and randomness introduced by different algorithms can render the interpretation of training curves less straightforward. Thus, we emphasize our primary focus on evaluation outcomes to provide a more comprehensive assessment.
>
>
> 2) In Section 3.4 (now Section 3.5), how many runs are used for these transfer experiments?
> A: Thank you for your inquiry. In the updated version, we conducted these transfer experiments a total of 20 times with different random seeds.  Notably, when transferring an easy-trained policy to a hard environment, the performance is notably inferior compared to the performance of the original hard-trained policy.
> However, a contrasting trend emerges when transferring a hard-trained policy to an easy environment. In this scenario, the performance does not deteriorate beyond the original performance of the easy-trained policy.
> Our results suggest that training policies in more challenging simulation environments, characterized by heightened noise and uncertainty, can yield policies that are more adaptable and capable of performing well when transferred to less challenging environments.
>
> [1] Duan, Yan, et al. "Benchmarking deep reinforcement learning for continuous control." International conference on machine learning. PMLR, 2016.
> [2] Laskin, Michael, et al. "URLB: Unsupervised Reinforcement Learning Benchmark." Deep RL Workshop NeurIPS 2021. 2021.
> [3] Welch, Bernard L. "The generalization of ‘STUDENT'S’problem when several different population varlances are involved." Biometrika 34.1-2 (1947): 28-35.

---

> > ### Author Response · Authors · 2023-08-07
> > **Response to Reviewer C8oN: Cont.**
> >
> > 3) Some experiments do not provide sufficient experimental details. For SAC, was the entropy scale hyperparameter swept?  Why were episodes cut off at 200 steps? Episode cutoffs are known to affect learning dynamics, with longer episodes typically being more difficult. How was convergence measured?
> > A: Thank you for your inquiries and observations.
> > For the SAC experiments, we utilized the automatic entropy coefficient update mechanism, as it is the default setting in the stable_baselines3 library. While we didn't explicitly mention the specific hyperparameter values swept for the entropy scale, we maintained the default settings provided by the library. To ensure transparency and facilitate reproducibility, we plan to make our code publicly available, which will provide a comprehensive view of the hyperparameters used.
> > Regarding the episode cutoff at 200 steps, our intention was to encourage rapid adaptation and stabilization of the OPS environment within a short period. In real-world scenarios, the optical system's state can become unpredictable and unstable without prompt control and stabilization. Furthermore, our focus was on learning efficient policies in simulation for subsequent transfer to real-world settings. In this context, shorter episodes can enhance the learning process by promoting quicker convergence.
> > In terms of measuring convergence, our concept of "convergence step" refers to the iteration count at which an algorithm reaches its peak performance. Beyond this point, further iterations result in marginal improvements, typically less than 0.1% of the previously achieved values.
> >
> > 4) Another note is that these reinforcement learning algorithms may not be fully tuned. The hyperparameters were chosen using 3 random seeds. It is well known that these algorithms exhibit large variations across random seeds.
> > A: Thank you for your valuable insight.
> > In the revised version of our paper, we conducted a grid search to identify optimal hyperparameters. Each set of hyperparameters was thoroughly evaluated using 5 different random seeds. While we acknowledge that even with these efforts, our chosen hyperparameters may not represent an absolute optimum, we believe that this approach aligns with contemporary benchmark practices [1].
> >
> > 5) Finally, is hard mode actually more difficult than medium mode of the OPS environment? Then in Figure 10 (a,b), transferring from training on medium mode to testing on hard mode results in similar final performance as training and testing on hard mode. Can the difficulty of hard mode be adjusted by adjusting the level of noise?
> > A: We sincerely appreciate your insightful observations and queries.
> > In the updated results, as depicted in Table 2, we confirm that the difficulty of the hard mode is notably higher than that of the medium mode. For a given algorithm, the performance within the hard mode significantly lags behind that of the medium mode.
> > This aligns with our intention to present a realistic representation of the varying complexities posed by different OPS modes.
> > Regarding the transfer experiments shown in Figure 10, we recognize the close performance of transferring a medium-trained model to the hard mode and the performance of an originally hard-trained model. This is an interesting finding, possibly attributed to the similarity between non-convex optimization problems posed by both the medium and hard modes. However, it's important to note that transferring from an easy-trained model to the hard environment yields worse results than training directly on the hard mode, emphasizing the distinction in difficulty levels. We will make the code publicly available, as it would allow researchers to manipulate the difficulty of the environment by adjusting noise levels and the initial point's randomness, providing a deeper understanding of its intricacies.

---

> > > ### Comment · Reviewer_C8oN · 2023-08-09
> > > **Reviewer Response**
> > >
> > > I thank the authors for their response, and especially for the inclusion of 10 more random seeds to support the claims made in the empirical study. I have adjusted my original comment accordingly.
> > >
> > > I did notice some additional minor issues that need fixing:
> > > - A number of in-text citations are not in parentheses
> > > - In the paragraph preceding Figure 6, I think that some of the text needs to be adjusted to reflect the new results of the full 20 runs. For example, TD3 does not converge to a value of -0.03 within 100,000 steps in the easy mode.
> > > - In Figures 9 and 10, are the shaded regions confidence intervals? Standard errors?
> > > - Y axis label of Figure 8(b) is cut off

---

> > > > ### Author Response · Authors · 2023-08-10
> > > > **Followup response**
> > > >
> > > > Thank you for pointing out these issues. We have addressed them in the revised version of the paper.  1) The citations have been adjusted to be in parentheses; 2) The text preceding Figure 6 has been updated to reflect the new results; 3) The Y-axis label of Figure 8(b) has been adjusted for visibility; 4) Description about the shaded regions in Figures 9 and 10 has been included to clarify that they represent standard deviations. Y

---

### Review · Reviewer_Pc1f · 2023-07-23

**Summary Of Contributions:**

This paper introduces a simulated RL environment for the application of controlling the timing of several light pulses to maximize the energy of the resulting pulse. They also provide initial benchmarking results of some of the most popular RL algorithms (TD3, SAC, PPO) on the environment in various configuration; these are also compared to the non-RL state-of-the-art stochastic parallel gradient descent (SPGD). Results suggest that RL algorithms may have improvements to offer the chosen application domain.

**Audience:**

Yes

**Broader Impact Concerns:**

Not knowing what pulse stacking is typically used for, I can't assess whether the application is of broader concern. Yet another reason to add a few motivating examples to the introduction.

**Claims And Evidence:**

Yes

**Requested Changes:**

Also see the weaknesses in the section above.

# Additional changes:
- The two paragraphs between Equations (4) and (5) are near duplicates.
- "If higher accuracy is required, one can include the noise term in the state [...], which transforms the control process into a POMDP." Can you elaborate on this? Why would this provide greater accuracy? Would the same algorithms still be appropriate then?
- Can you discuss the actual wall time of these experiments? How practical is it to run these experiments for 300k steps? Would a practitioner train a policy in simulation and load a frozen policy into a physical system? If not, is it possible to give realtime state and reward feedback at low enough latency?

**Strengths And Weaknesses:**

# Caveat:

I'm approaching this work as an RL practitioner with a Physics undergraduate degree. As such my understanding of the optical system that is targeted by this simulation environment may be limited and incomplete and so, too, my knowledge of the relevant optics literature.

# Strengths:
- Simple and clear writing.
- Straightforward application of RL to a novel application domain.

# Weaknesses:
- Motivation for the optical stacking is missing. Why do people want to stack these signals in practice?
- Surprising failure of PPO in the easy setting in Fig 6(a). A good discussion of why would be of interest to the RL community.
- Discussion of limitations missing. While there is configurable noise in the state transition, is this the only source of added challenge in reality? A discussion of the ways the environment still falls short of realism (and could be improved in future) is missing. One question in particular that came to mind immediately is whether it is realistic to assume the algorithm has access to the next state instantaneously (without hardware-dependent delay?) or whether the noise in Equation (4) is enough to incorporate this systematic delay.

---

> ### Author Response · Authors · 2023-08-07
> **Response to Reviewer Pc1f**
>
> 1) Motivation for the optical stacking is missing. Why do people want to stack these signals in practice?
> A: Thank you for your insightful comment. We have incorporated a discussion on the motivation behind optical stacking in the revised manuscript. This is detailed in the new Appendix A.1.
> As explained there, the practical need for higher pulse energy is a driving factor for exploring methods like Optical (coherent) pulse stacking. This approach offers a promising and straightforward way to scale pulse energy, making it applicable in various fields, including industry and scientific research.
>
> 2) Surprising failure of PPO in the easy setting in Fig 6(a).
> Thank you for your thought-provoking observation.  We have elaborated on this discussion in Appendix B.5 and provided a concise overview in section 3.2.
> As an on-policy algorithm, PPO's operational principle hinges on recent data for policy enhancement. When applied to our OPS environment, which harbors multiple local optima (as indicated in Figure 2), PPO's exploration limitations and preference for recent data can result in getting stuck within local optima, hampering its ability to unearth globally optimal policies.
> In contrast, SAC and TD3 leverage historical experiences from a replay buffer, enabling them to transcend the confines of recent data. This superior data utilization generally culminates in enhanced convergence and potentially superior solutions compared to on-policy methods.
> Our findings harmonize with those from a comprehensive benchmark study [1]. As exemplified in Figures 6 and 8 of [2], SAC consistently succeeds in resolving all posed tasks, whereas PPO demonstrates competence in most scenarios but struggles in certain instances.
>
> 3) Discussion of limitations missing. While there is configurable noise in the state transition, is this the only source of added challenge in reality? One question in particular that came to mind immediately is whether it is realistic to assume the algorithm has access to the next state instantaneously (without hardware-dependent delay?)
> A: Thank you for raising a crucial point.   We have incorporated this limitation discussion in section 4.3 of the revised manuscript.
> In the actual OPS system, we encounter two distinct types of noise. The first arises from temperature drift, resulting from the thermal expansion of mirrors or delay line devices. This type of noise introduces gradual, slow-varying perturbations. Our simulation models this by employing a space-wise linear function, mirroring empirical observations detailed in [2].
> The second variety is fast noise due to the vibration of delay line devices. We approximate this rapid noise as Gaussian noise in our experimentation. However, it's crucial to recognize that real-world vibration-induced noise may not always conform to a Gaussian distribution. The noise attributes could be influenced by factors such as vibration properties, symmetry, or non-linear effects. While Gaussian noise is often a reasonable approximation as per the Central Limit Theorem, it might not accurately capture all nuances of real-world vibrations.
> A notable distinction between our simulations and real experiments is the divergence in noise properties. Future research aims to address these discrepancies and bridge the gap between simulations and reality. Our ongoing efforts involve exploring sim2real transfer strategies and refining the simulation to better match real-world noise characteristics.
> Regarding your question about feedback delay (or hardware-dependent delay), we've replicated this aspect in our simulation by introducing free-running noise on delay lines, simulating the process of photo-detection, data preprocessing, and action computation. We have elaborated on this implementation in Appendix B.6 and conducted additional experiments to assess policy transfer across different feedback levels.
>
>
> [1] Yu, Tianhe, et al. "Meta-world: A benchmark and evaluation for multi-task and meta reinforcement learning." Conference on robot learning. PMLR, 2020.
> [2] Kodet, Jan, and Ivan Prochazka. "Note: Optical trigger device with sub-picosecond timing jitter and stability." Review of scientific instruments 83.3 (2012).

---

> > ### Author Response · Authors · 2023-08-07
> > **Response to Reviewer Pc1f: Cont.**
> >
> > 4) The two paragraphs between Equations (4) and (5) are near duplicates.
> > A: Thank you for bringing this to our attention. We have addressed the issue of duplicated text between equations (4) and (5).
> >
> > 5) "If higher accuracy is required, one can include the noise term in the state [...], which transforms the control process into a POMDP." Can you elaborate on this? Why would this provide greater accuracy? Would the same algorithms still be appropriate then?
> > A: Thank you for your question.  In the context of our OPS system, the presence of time-dependent noise, specifically temperature drift, introduces a non-Markovian aspect. This is because the subsequent state at time step $t+1$, labeled as a state $s_{t+1}$, is influenced not only by the current state $s_{t}$ and the action $a_{t}$ taken, but also by the specific time $t$. Such temporal dependency deviates from the Markov property, which can potentially hinder the performance of RL algorithms designed for Markov Decision Processes (MDPs), like PPO, SAC, and TD3.
> > By introducing the noise factor $e_t$ into the state definition, we augment a next state  $\hat{s_{t+1}}= [s_{t+1}, e_{t+1}]$  that exclusively relies on the present state $\hat{s_{t}} = [s_t, e_t] $ and the action $a_t$.
> > This transformation leads to a scenario known as a Partially Observable Markov Decision Process (POMDP), where the noise vector $e_t$ remains partly unobservable. POMDPs open up the utilization of various RL algorithms specifically designed for such scenarios, including methods like Soft Actor-Critic for POMDPs [3]. These techniques have the potential to provide improved outcomes.
> > However, it's important to note that, in the scope of this work, our primary focus remains on the simulation environment and general RL benchmarks. As a result, we haven't extensively explored algorithms tailored exclusively for POMDPs. Nevertheless, our experimentation with standard MDP-based RL algorithms, such as SAC, has yielded noteworthy performance outcomes.
> > To ensure precision, I've rephrased the original claim as: "By incorporating the noise term into the state definition as $\hat{s_t} = [s_t, e_t]$, we effectively convert the control process into a Partially Observable Markov Decision Process (POMDP)."
> >
> > 6) Can you discuss the actual wall time of these experiments? How practical is it to run these experiments for 300k steps? Would a practitioner train a policy in simulation and load a frozen policy into a physical system? If not, is it possible to give realtime state and reward feedback at low enough latency?
> > A:  We sincerely appreciate your invaluable suggestions. We have incorporated the runtime analysis in Section 3.3, along with relevant discussions in Section 4.1. Given the substantial temporal and energy expenses, coupled with the necessity for expert intervention in the manual calibration of optical devices, training the RL policy in a simulation and subsequently transferring it to real-world experiments emerges as a judicious approach.
> >
> > [3]  Wahid, Ayzaan, et al. "Learning object-conditioned exploration using distributed soft actor critic." Conference on Robot Learning. PMLR, 2021

---

### Review · Reviewer_cc9b · 2023-07-27

**Summary Of Contributions:**

The manuscript introduces a control problem arising in optical setups that are based on the goal of stacking several optical pulses generated by a laser. The overall goal of the task is to optimize the total energy of the stacked pulse by controlling different delay lengths in a stacking stage. This pulse stacking problem exhibits various complexities, including slow time-dependent noise and fast-varying noise, multiple local minima, and a non-convex error function. Further, scaling the number of to-be-stacked pulsed effectively increases the search space, the number of local minima and accordingly the difficulty. After introducing the problem, the authors propose an open-source simulation environment that emulates the optical experiment allowing for fast iteration and benchmarking of different control algorithms. In the following, different algorithms are evaluated in this environment. The authors hereby focus on four methods known from Reinforcement Learning (RL) that are evaluated in this simulation environment to solve the task, namely SPGD, PPO, SAC, and TD3. Furthermore, different versions of the task are presented that are generated by considering different varying noise conditions, initialization, and the number of stacked pulses considered, resulting in tasks of varying difficulty levels. The difficulty levels claimed are verified by evaluating the different algorithms and finding clear increasing errors and longer convergence times towards more difficult problems. Thereby, the results demonstrate that the easiest task, where the system is already close to a minimum, is effectively solved by SPGD, while tasks with more complex noise conditions require more advanced methods. Among the tested approaches, TD3 emerges as the most promising method for solving the pulse stacking task effectively.

**Audience:**

Yes

**Broader Impact Concerns:**

There are no concerns on the ethical implications of this work

**Claims And Evidence:**

Yes

**Requested Changes:**

# Main Requests
- add justification/validation that the simulation environment is close to the experimental setup
- move the chosen parameters/model sizes as presented in the appendix toward the main text and discuss the influence of model sizes
- add the time it takes to train/evaluate the different RL methods
- add a discussion on potential reasons why some of the methods work better than others

# Minor Requests
- remove doubled text in between equations (4) and (5) on page 5 + remove typos in the manuscript

**Strengths And Weaknesses:**

The optical pulse stacking problem is presented clearly and the performance evaluations of the RL methods in the simulation environment seem rigorous. Further, the proposed simulation environment allows for scalable benchmarking, in particular, due to the adaptable dimension of search space and different noise settings. As the authors claim their results will attract readers from the ML and optics community, however, the RL methods are presented very briefly making it hard to understand their difference for people from the optics community. Additionally, the manuscript focuses on accuracy as the main metric. In the automation of experimental setups, also other metrics such as execution time, required resources, etc. of the algorithms are important. Finally, the authors claim that their simulation environment is close to real-world experiments, however, besides showing the experimental setup in the appendix, there is no direct comparison of the behavior of the simulation environment and the real experiment. Accordingly, this claim seems not well justified and it is unclear how the model trained in the simulation environment would perform in the real-world experiment.

# Strengths
- the manuscript raises important points regarding the automation of the adjustment of optical setups
- the authors propose a scalable open-source environment to evaluate and compare different RL methods on an optical control problem
- in the simulation environment, the problem can be decomposed into different difficulty levels, this might help to analyze and evaluate different algorithms

# Weaknesses
- there is no comparison of the simulation environment towards the real experiment, it would have been great to see how the simulation-based trained policies perform in controlling the real-world experiment and how close the simulations are to the experiment
- the manuscript does not explain well the used methods, for people from the optics community that might not be fully confirmed with the ML methods this might be hard to follow
- the authors focus mainly on performance as the metric and do not provide how long the training of the different methods takes (e.g. on a standard CPU/GPU). However, this might be an important consideration for people that have to solve real-world problems where time is always an important resource
- after evaluating the methods the authors do not discuss why the methods lead to different performances. Can they identify certain features of the algorithm that seem to be important for solving the tasks?

---

> ### Author Response · Authors · 2023-08-07
> **Response to Reviewer cc9b**
>
> 1) add justification/validation that the simulation environment is close to the experimental setup.
> A:  Thank you for your comments. We have included a related discussion within Section 4.1 of the revised manuscript.
> While we did not include a direct comparison between the simulation and real-world experiments in this paper, we want to emphasize that the correctness and accuracy of our simulation environment have been partially validated in previous works, specifically in [1].
> In [1], a simplified version of our OPS environment, focusing on the 1-stage OPS system, was subjected to both simulations and real-world experimentation to assess the performance of RL algorithms. The outcomes of this study indicated that RL algorithms exhibited effectiveness in both the simulated and real settings, validating the usefulness of simulations as a training tool and their potential for saving valuable time and resources compared to real-world trials.
> While our current OPS simulation and experimental configurations are more intricate than those presented in [1], the fundamental physical principles remain consistent. This alignment provides us with a solid foundation for confidence in the reliability of our simulation environment.
>
> 2) add the time it takes to train/evaluate the different RL methods.
> A: Thank you for your valuable suggestions. We have included a comprehensive running time analysis in Section 3.3 of the revised manuscript.
>
> 3) add a discussion on potential reasons why some of the methods work better than others.
> A: Thank you for your insightful feedback. We have detailed this discussion in Appendix B.5 and provided a succinct overview in section 3.2.
> Our experiments showcased a distinct performance advantage of SAC and TD3 over PPO.
> As an on-policy algorithm, PPO heavily relies on recent experiences for policy updates [2]. In the context of our OPS environment, replete with multiple local optimal points (as evidenced in Figure 2), PPO's exploration constraints and preference for recent data can lead to confinement within local optima, thereby limiting its potential to unearth globally optimal policies.
> In contrast, off-policy algorithms,  SAC and TD3 can draw on historical experiences stored in a replay buffer, allowing them to break free from the constraints of recent data. This enhanced data utilization generally leads to improved convergence and potentially better solutions compared to on-policy methods.
> Our findings align with insights from a comprehensive benchmark study [3]. As showcased in Figures 6 and 8 of [3], SAC consistently succeeds in solving all presented tasks, while PPO demonstrates proficiency in most scenarios but falters in select cases.
>
> 4) move the chosen parameters/model sizes as presented in the appendix toward the main text and discuss the influence of model sizes.
> A: Thank you for your valuable feedback. We have added a discussion about the model size and its influence in appendix B.2. While we recognize the importance of this aspect, our main text is subject to page limitations.
>
> 5) the manuscript does not explain well the used methods, for people from the optics community that might not be fully confirmed with the ML methods this might be hard to follow.
> A: Thank you for your suggestions.  In the revision, we have included the illustration of RL algorithms in Appendix B.1 for clarity and completeness.
>
> 6) remove doubled text in between equations (4) and (5) on page 5 + remove typos in the manuscript.
> A: Thank you for bringing this to our attention. We have addressed the issue of duplicated text between equations (4) and (5).
>
> [1] H. Tünnermann et al. “Deep reinforcement learning for coherent beam combining applications.” Opt. Express, 2019.
> [2] Sutton, Richard S., and Andrew G. Barto. Reinforcement learning: An introduction. MIT press, 2018.
> [3] Yu, Tianhe, et al. "Meta-world: A benchmark and evaluation for multi-task and meta reinforcement learning." Conference on robot learning. PMLR, 2020.

---

### Author Response · Authors · 2023-08-07
**Revision of the submission**

We extend our sincere gratitude to the reviewers for their invaluable insights.
We have revised our manuscript in accordance with the feedback received. Within the manuscript, we have distinguished the revised sections by employing colored. The principal modifications are summarized as follows:
1. To enhance the robustness and credibility of our findings, we have conducted experiments across 20 different random seeds. Moreover, for important outcomes, including the final pulse energy in the evaluation, we have introduced Welch’s t-test as a statistical validation mechanism, substantiating the discernible disparities between various algorithms.
2. In the revised manuscript, we have integrated a running time analysis within Section 3.3, contributing a comprehensive perspective.
3. We have introduced a discussion regarding the suboptimal performance of PPO in our experiments, encapsulated within Appendix B.5.

---

### Decision · Action_Editors · 2023-09-24

**Recommendation:** Accept as is

**Comment:**

The paper introduces a new reinforcement learning environment that is motivated by an application in Optics (Optical Pulse Stacking). We had expert reviewers from both RL and the Optics communities. They have raised several issues in their initial reviews, including the statistical significance of some of the results.
The authors diligently revised the paper to address them. The reviewers are now all in favour of accepting the paper (Two Accepts and one Leaning Accept). They note that the paper might be of interest to both ML and optics communities. I recommend acceptance of this paper.

**Audience:**

The paper is of interest to the reinforcement learning community, as it introduces a new benchmarking environment, as well as to the Optics community, as it shows the feasibility of RL algorithms to solve a problem in optics.

**Claims And Evidence:**

Yes, the paper adequately supports its claims.